# OGC: Unsupervised 3D Object Segmentation from Rigid Dynamics of Point Clouds

**Ziyang Song**    **Bo Yang**

vLAR Group, The Hong Kong Polytechnic University
ziyang.song@connect.polyu.hk    bo.yang@polyu.edu.hk

## Abstract

In this paper, we study the problem of 3D object segmentation from raw point clouds. Unlike all existing methods which usually require a large amount of human annotations for full supervision, we propose the first unsupervised method, called OGC, to simultaneously identify multiple 3D objects in a single forward pass, without needing any type of human annotations. The key to our approach is to fully leverage the dynamic motion patterns over sequential point clouds as supervision signals to automatically discover rigid objects. Our method consists of three major components, 1) the object segmentation network to directly estimate multi-object masks from a single point cloud frame, 2) the auxiliary self-supervised scene flow estimator, and 3) our core object geometry consistency component. By carefully designing a series of loss functions, we effectively take into account the multi-object rigid consistency and the object shape invariance in both temporal and spatial scales. This allows our method to truly discover the object geometry even in the absence of annotations. We extensively evaluate our method on five datasets, demonstrating the superior performance for object part instance segmentation and general object segmentation in both indoor and the challenging outdoor scenarios. Our code and data are available at https://github.com/vLAR-group/OGC

## 1 Introduction

Identifying 3D objects from point clouds is vital for machines to tackle high-level tasks such as autonomous planning and manipulation in real-world scenarios. Inspired by the seminal work PointNet [56], a plethora of sophisticated models [70; 76; 37] have been developed to accurately detect and segment individual objects from the sparse and irregular point clouds. Although these methods have achieved excellent performance on a wide range of public datasets, they primarily rely on a huge amount of human annotations for full supervision. However, it is extremely costly to fully annotate every objects in point clouds due to the irregularity of data format.

Very recently, a few works start to address 3D object segmentation in the absence of human annotations. By analysing 3D scene flows from sequential point clouds, Jiang *et al*. [28] apply the conventional subspace clustering optimization technique to identify moving objects from raw point cloud sequences. With the self-supervised learning of 3D scene flow, SLIM [3] is the first learning-based work to showcase that the set of moving points can be effectively learned as an object against the stationary background. Fundamentally, their design principle shares the key spirit of Gestalt theory [73; 69] developed exactly 100 years ago: the raw sensory data with similar motion are likely to be organized into a single object. This is indeed true in the real world where solid objects usually have strong correlation in rigid motions. However, these methods cannot learn to simultaneously segment multiple interested 3D objects from a single point cloud in one go.

36th Conference on Neural Information Processing Systems (NeurIPS 2022).

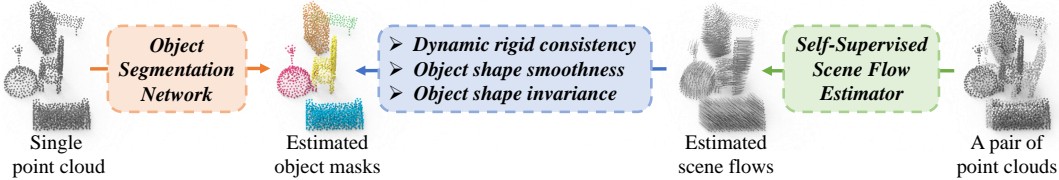

Figure 1: The general workflow and components of our framework.

Motivated by the potential of motion dynamics, this paper aims to design a general neural framework to simultaneously segment multiple 3D objects, without requiring any human annotations but the inherent object dynamics in training. To achieve this, a naïve approach is to train a neural network to directly cluster motion vectors into groups from sequential point clouds, which is widely known as motion segmentation [77; 78]. However, such design requires that the input data points are sequential in both training and testing phases, and the trained model cannot infer objects from a single point cloud. Fundamentally, this is because the learned motion segmentation strategies simply cluster similar motion vectors instead of discriminating object geometries, and therefore such design is not general enough for real-world applications.

In this regard, we design a new pipeline which takes a single point cloud as input and directly estimates multiple object masks in a single forward pass. Without needing any human annotations, our pipeline instead leverages the underlying dynamics of sequential point clouds as supervision signals. In particular, as shown in Figure 1, our architecture consists of three major components: 1) an object segmentation network to extract per-point features and estimate all object masks from *a single point cloud*, as indicated by the orange block; 2) an auxiliary self-supervised network to estimate per-point motion vectors from a pair of point clouds, as indicated by the green block; 3) a series of loss functions to fully utilize the motion dynamics to supervise the object segmentation backbone, as indicated by the blue block. For the first two components, it is actually flexible to adopt any of existing neural feature extractors [57] and self-supervised motion estimators [34]. Nevertheless, the third component is particularly challenging to design, primarily because we need to take into account not only the consistency of diverse dynamics of multiple objects in a sequence, but also the invariance of object geometry irregardless of different moving patterns.

To tackle this challenge, we introduce three key losses to end-to-end train our object segmentation network from scratch: 1) a multi-object dynamic rigid consistency loss, which aims to evaluate how coherently all estimated object masks (shapes) can fit the motion via rigid transformations; 2) an object shape smoothness prior, which regularizes all points of each estimated object to be spatially continual instead of fragmented; 3) an object shape invariance loss, which drives multiple estimated masks of a particular object to be invariant given different (augmented) rigid transformations. These losses together force all estimated **o**bjects' **g**eometry to be **c**onsistent and represented by high-quality masks, purely from raw 3D point clouds without any human annotations. Our method is called **OGC** and our contributions are:

- We introduce the first unsupervised multi-object segmentation pipeline on single point cloud frames, without needing any human annotations in training or multiple frames as input.
- We design a set of geometry consistency based losses to fully leverage the object rigid dynamics and shape invariance as effective supervision signals.
- We demonstrate promising object segmentation performance on five datasets, showing significantly better results than classical clustering and optimization baselines.

**Difference from Scene Flow Estimation:** We do not aim to design a new scene flow estimation method such as [74; 34; 20; 71]. Instead, we use unsupervised learning based per-point scene flow as supervision signals for single-frame multi-object segmentation.

**Difference from Motion Segmentation:** We neither aim to segment motion vectors such as [71; 63] which require multiple successive frames as input in both training and testing. Instead, our network directly estimates object masks from single frames, and therefore is more flexible and general.

**Scope:** This paper does not intend to replace fully-supervised approaches because the never-moving objects are unlikely to be discovered due to the lack of supervision signals. In addition, estimating object categories or non-rigid objects such as articulated buses and semi-truck with trailers is also out of the scope of this paper.

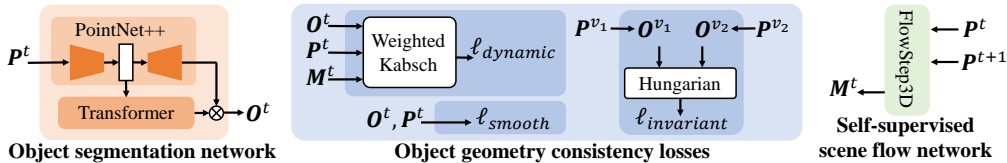

Figure 2: Components of our pipeline. The object segmentation network consists of PointNet++ and Transformer decoders. FlowStep3D is adopted as the self-supervised scene flow network.

## 2 Related Works

**Fully-supervised 3D Object Segmentation:** To identify 3D objects from point clouds, existing fully-supervised solutions can be divided as 1) bounding box based object detection methods [84; 37; 60] or 2) mask based instance segmentation pipelines [70; 76; 68]. Thanks to the dense human annotations and the well developed backbones including projection-based [38; 8; 37], point-based [57; 65; 26] and voxel-based [22; 12] feature extractors, these methods achieve impressive performance on both indoor and outdoor datasets. However, manually labelling every object in large-scale point clouds is costly. To alleviate this burden, we aim to pioneer 3D object segmentation without human labels.

**3D Scene Flow Estimation and Motion Segmentation:** Given sequential point clouds, per-point 3D motion vectors, also known as scene flow, can be accurately estimated. Early works focus on fully-supervised scene flow estimation [42; 4; 24; 43; 55; 53; 72], whereas recent methods start to explore self-supervised motion estimation [50; 74; 34; 46; 82; 39]. Taking the scene flow as input, a number of works [80; 27; 64; 3] aim to group similar motion vectors, and then obtain bounding boxes or masks only for dynamic objects. Although achieving encouraging results, they either rely on ground truth segmentation for supervision or can only segment simple foreground and background objects, without being able to simultaneously segment multiple objects. In this paper, we leverage the successful self-supervised scene flow estimator as our auxiliary neural network to provide valuable supervision signals, so that multiple objects can be identified in a single forward pass.

**Unsupervised 2D Object Segmentation:** Inspired by the early work AIR [16], a large number of generative models have been proposed to discover objects from single images without needing human annotations, including MONet [6], IODINE [23], Slot-Att [44], *etc.*. These methods are further extended to segment objects from video frames [35; 25; 49; 29; 85; 15]. However, as investigated by the recent work [79], all these approaches can only process simple synthetic datasets, and cannot discover objects from complex real-world images yet. It is still elusive to apply these ideas on 3D point clouds where 3D objects are far more complicated and diverse in terms of geometry.

**2D Scene Flow Estimation and Motion Segmentation:** Given image sequences, pixel-level 2D scene flow, also known as optical flow, have been extensively studied in literature [18; 83]. The estimated flow field can be further grouped as objects [61; 62; 10; 45; 77; 41; 32]. Drawing insights from these works, this paper aims to segment multiple diverse objects in the complex 3D space.

## 3 OGC

### 3.1 Overview

As shown in Figure 2, given a single point cloud $\boldsymbol{P}^t$ with $N$ points as input, *i.e.*, $\boldsymbol{P}^t \in \mathbb{R}^{N \times 3}$, where each point only has a location $\{x, y, z\}$ without color for simplicity, the **object segmentation network** extracts per-point features and directly reasons a set of object masks, denoted as $\boldsymbol{O}^t \in (0, 1)^{N \times K}$, where $K$ is a predefined number of objects that is large enough for a specific dataset. In particular, we firstly adopt PointNet++ [57] to extract the per-point local features. Then we employ Transformer decoders [67] to attend to the point features and yield all object masks in parallel. The whole architecture can be regarded as a 3D extension of the recent MaskFormer [9] which shows excellent performance in object segmentation in 2D images. Thanks to the powerful Transformer module, each inferred object mask is effectively modeled over the entire point cloud. Implementation details are in Appendix A.1

In the meantime, we have the corresponding sequence of point clouds for supervision, denoted as $\{\boldsymbol{P}^t, \boldsymbol{P}^{t+1}, \cdots\}$. For simplicity, we only use the first two frames $\{\boldsymbol{P}^t, \boldsymbol{P}^{t+1}\}$ and feed them into

the **auxiliary self-supervised scene flow network**, obtaining satisfactory motion vectors for every point in the first point cloud frame, denoted as $\boldsymbol{M}^t \in \mathbb{R}^{N \times 3}$, where each motion vector represents point displacement $\{\Delta x, \Delta y, \Delta z\}$. Among the existing self-supervised scene flow methods, we choose the recent FlowStep3D [34] which shows excellent scene flow estimation in multiple datasets. Implementation details are in Appendix A.1. To train the object segmentation network from scratch, the key component is the supervision mechanism as discussed below.

### 3.2 Object Geometry Consistency Losses

Given the input point cloud $\boldsymbol{P}^t$ and its output object masks $\boldsymbol{O}^t$ and motion $\boldsymbol{M}^t$, we introduce the following objectives to satisfy the geometry consistency on both frames $\boldsymbol{P}^t$ and $(\boldsymbol{P}^t + \boldsymbol{M}^t)$. Note that, the masks $\boldsymbol{O}^t$ are meaningless at the very beginning and need to be optimized appropriately.

#### (1) Geometry Consistency over Dynamic Object Transformations

From time $t$ to $t+1$, the rigid objects in point cloud frame $\boldsymbol{P}^t$ usually exhibit different dynamic transformations which can be described by matrices belonging to $SE(3)$ group. For the $k^{th}$ object, we firstly retrieve its (soft) binary mask $\boldsymbol{O}_k^t$, and then feed the tuple $\{\boldsymbol{P}^t, \boldsymbol{P}^t + \boldsymbol{M}^t, \boldsymbol{O}_k^t\}$ into the differentiable weighted-Kabsch algorithm [31; 21], estimating its transformation matrix $\boldsymbol{T}_k \in \mathbb{R}^{4 \times 4}$.

In order to drive all raw object masks to be more and more accurate, so as to fully explain the corresponding motion patterns within all masks, the following dynamic rigid loss is designed to minimize the discrepancy of per-point scene flow between time $t$ and $t+1$ for each point in $\boldsymbol{P}^t$:

$$\ell_{dynamic} = \frac{1}{N} \sum_{n=1}^{N} \left\| \left( \sum_{k=1}^{K} o_{nk} \cdot (\boldsymbol{T}_k \circ \boldsymbol{p}_n) \right) - (\boldsymbol{p}_n + \boldsymbol{m}_n) \right\|_2 \tag{1}$$

where $o_{nk} \in (0, 1)$ represents the probability of being assigned to the $k^{th}$ object for a specific $n^{th}$ point $\boldsymbol{p}_n$, and $\boldsymbol{m}_n \in \mathbb{R}^3$ represents the motion of $\boldsymbol{p}_n$. The operation $\circ$ applys the rigid transformation to the point. Intuitively, if one inferred object mask includes two sets of points with two different moving directions, the transformed point cloud can only favor one moving direction, thereby resulting in higher errors. Therefore, $\ell_{dynamic}$ can push all object masks to fit the dynamic and diverse motion patterns. However, here arises a critical issue: a single rigid object may be assigned to multiple masks, *i.e.* oversegmentation. We alleviate this issue by a simple smoothness regularizer discussed below.

We observe that such rigid constraint concept is also applied in recent scene flow estimation method [20]. However, their objective is to push the scene flow to be consistent given object masks (estimated by DBSCAN clustering), while our objective is to learn high-quality masks from given flows.

#### (2) Geometry Smoothness Regularization

The primary reason why a single object may be oversegmented is the lack of spatial connectivity between individual points. However, our common observation is that physically neighbouring points usually belong to a single object. In this regard, we simply introduce a geometry smoothness regularizer. Particularly, for a specific $n^{th}$ 3D point $\boldsymbol{p}_n$ in the point cloud $\boldsymbol{P}^t$, we firstly search $H$ points from its neighbourhood using either KNN or spherical querying methods, and then force their mask assignments to be consistent with the center point $\boldsymbol{p}_n$. Mathmatically, it is defined as:

$$\ell_{smooth} = \frac{1}{N} \sum_{n=1}^{N} \left( \frac{1}{H} \sum_{h=1}^{H} d(\boldsymbol{o}_n, \boldsymbol{o}_{n_h}) \right) \tag{2}$$

where $\boldsymbol{o}_n \in (0, 1)^K$ represents the object assignment of center point $\boldsymbol{p}_n$, and $\boldsymbol{o}_{n_h} \in (0, 1)^K$ represents the object assignment of its $h^{th}$ neighbouring point. The distance function $d()$ is flexible to choose $L1$ / $L2$ or a more aggressive cross-entropy function.

Note that, such local smoothness prior is successfully used for scene flow estimation [42; 34]. Here, we instead demonstrate its effectiveness for object segmentation.

#### (3) Geometry Invariance over Scene Transformations

With the above geometry constraints designed in (1)(2), the shapes of dynamic objects can be reasonably segmented. However, the learned object geometry may not be general enough. For example, a moving car can be well segmented, yet another similar parked car may not be discovered. To this end, we introduce an object geometry invariance constraint as follows:

- Firstly, given $P^t$, we apply two transformations to get augmented point clouds $P^{v_1}$ and $P^{v_2}$.
- Secondly, we feed $P^{v_1}$ and $P^{v_2}$ into our object segmentation network, obtaining two sets of object masks $O^{v_1}$ and $O^{v_2}$. Because the per-point locations in two point clouds are transformed differently, the position sensitive PointNet++ [57] features generate two different sets of masks.
- Thirdly, we leverage the Hungarian algorithm [36] to one-one match the individual masks in $O^{v_1}$ and $O^{v_2}$ according to the object pair-wise IoU scores. Basically, this is to address the issue that there is no fixed order for predicted object masks from the two augmented point clouds.
- At last, we reorder the masks in $O^{v_2}$ to align with $O^{v_1}$, and design the invariance loss as follows.

$$\ell_{invariant} = \frac{1}{N} \sum_{n=1}^{N} \hat{d}\big(\hat{\boldsymbol{o}}_n^{v_1}, \hat{\boldsymbol{o}}_n^{v_2}\big) \tag{3}$$

where $\hat{\boldsymbol{o}}_n^{v_1}$ and $\hat{\boldsymbol{o}}_n^{v_2}$ are the reordered object assignments of the two augmented point clouds for the $n^{th}$ point. The distance function $\hat{d}()$ is flexible to use $L1$, $L2$ or cross-entropy. Ultimately, this loss drives the estimated object masks to be invariant with different views of input point clouds.

Notably, unlike existing self-supervised learning [11] which usually uses invariance prior for better latent representations, here we aim to generalize the segmentation strategy to similar yet static objects.

### 3.3 Iterative Optimization of Object Segmentation and Motion Estimation

With the designed geometry consistency loss functions, the object segmentation network is optimized from scratch by the combined loss: $\ell_{seg} = \ell_{dynamic} + \ell_{smooth} + \ell_{invariant}$. For efficiency, the auxiliary self-supervised scene flow network FlowStep3D [34] is independently trained by its own losses until convergence. Intuitively, with better and better object masks estimated, the estimated scene flow is also expected to be improved further if we use the masks properly. To this end, we propose the following Algorithm 1 to iteratively improve object segmentation and motion estimation.

---

**Algorithm 1** Iterative optimization of object segmentation and scene flow estimation. Assume the whole train split has $S$ point cloud pairs: $\{(P^t, P^{t+1})_1 \cdots (P^t, P^{t+1})_S\}$.

---

*Stage 0: Initial scene flow estimation.*
  • Independently and fully train the self-supervised scene flow network on the whole training data split, and obtain reasonable scene flow estimations: $\{(P^t, P^{t+1}, M^t)_1 \cdots (P^t, P^{t+1}, M^t)_S\}$.
**for** number of iteration rounds $R$ **do**
  *Stage 1: Object segmentation optimization.*
  • Train the object segmentation network using $\ell_{seg}$ for a total $E$ epochs on the whole training split: $\{(P^t, P^{t+1}, M^t)_1 \cdots (P^t, P^{t+1}, M^t)_S\}$.
  • Estimate reasonable object masks: $\{(P^t, P^{t+1}, O^t, O^{t+1})_1 \cdots (P^t, P^{t+1}, O^t, O^{t+1})_S\}$.
  *Stage 2: Scene flow improvement.*
  • For each pair of data $(P^t, P^{t+1}, M^t, O^t, O^{t+1})$, by drawing insights from the classical ICP [2], we propose an **object-aware ICP** algorithm to estimate new scene flow $\hat{M}^t$ for point cloud $P^t$.
  • Update the new scene flow for next round training:
$\{(P^t, P^{t+1}, M^t)_1 \cdots (P^t, P^{t+1}, M^t)_S\} \leftarrow \{(P^t, P^{t+1}, \hat{M}^t)_1 \cdots (P^t, P^{t+1}, \hat{M}^t)_S\}$

---

Empirically, setting the total number of rounds $R$ to be 2 or 3 has a good trade off between accuracy and training efficiency. Due to the limited space, details of object-aware ICP algorithm are in Appendix A.2. We exclude the invariance loss $l_{invariant}$ from object segmentation optimization stage in the early rounds so that the networks can focus on moving objects in training and produce better scene flows, and then add $l_{invariant}$ back in the final round. Detailed analysis is in Appendix A.5.

## 4 Experiments

Our method is evaluated on four different application scenarios: 1) part instance segmentation of articulated objects on SAPIEN dataset [75], 2) object segmentation of indoor scenes on our own synthetic dataset, 3) object segmentation of real-world outdoor scenes on KITTI-SF dataset [48], and 4) object segmentation on the sparse yet large-scale LiDAR based KITTI-Det [19] and SemanticKITTI [5] datasets. For evaluation metrics, we follow [51] and report the **F1-score**, **Precision**, and **Recall**

Table 1: Quantitative results of our method and baselines on the SAPIEN dataset.

| | | AP↑ | PQ↑ | F1↑ | Pre↑ | Rec↑ | mIoU↑ | RI↑ |
|---|---|---|---|---|---|---|---|---|
| Supervised Methods | PointNet++ [57] | - | - | - | - | - | 51.2 | 65.0 |
| | MeteorNet [43] | - | - | - | - | - | 45.7 | 60.0 |
| | DeepPart [80] | - | - | - | - | - | 53.0 | 67.0 |
| | MBS [27] | - | - | - | - | - | 67.3 | 77.0 |
| | OGC$_{sup}$ | 66.1 | 48.7 | 62.0 | 54.6 | 71.7 | 66.8 | 77.1 |
| Unsupervised Motion Segmentation | TrajAffn [52] | 6.2 | 14.7 | 22.0 | 16.3 | 34.0 | 45.7 | 60.1 |
| | SSC [51] | 9.5 | 20.4 | 28.2 | 20.9 | 43.5 | 50.6 | 65.9 |
| Unsupervised Methods | WardLinkage [30] | 17.4 | 26.8 | 40.1 | 36.9 | 43.9 | 49.4 | 62.2 |
| | DBSCAN [17] | 6.3 | 13.4 | 20.4 | 13.9 | 37.9 | 34.2 | 51.4 |
| | **OGC(Ours)** | **55.6** | **50.6** | **65.1** | **65.0** | **65.2** | **60.9** | **73.4** |

with an IoU threshold of 0.5. In addition, we report the Average Precision (**AP**) score following COCO dataset [40] and the Panoptic Quality (**PQ**) score defined in [33]. The mean Intersection over Union (**mIoU**) score and the Rand Index (**RI**) score implemented in [27] are also included. Note that, all metrics are computed in a class-agnostic manner.

### 4.1 Evaluation on SAPIEN Dataset

The SAPIEN dataset [75] provides 720 simulated articulated objects with part instance level annotations. Each object has 4 sequential scans. The part instances have different articulating (moving) states. We follow [27] to use the training data generated from [81]. In particular, there are 82092 pairs of point clouds for training, 2880 single point cloud frames for testing. Each point cloud is downsampled to 512 points in both training and testing.

Since there is no existing unsupervised method for multi-object segmentation on 3D point clouds, we firstly implement two classical clustering methods: WardLinkage [30] and DBSCAN [17] to directly group 3D points from single point clouds into objects. Secondly, we implement two classical motion segmentation methods: TrajAffn [52] and SSC [51]. Note that, these two methods take the same estimated scene flows of FlowStep3D as input, while our method uses the estimated scene flows during training only, but takes single point clouds as input during testing. In addition, we also include the excellent results of several fully-supervised methods (PointNet++ [57], MeteorNet [43], DeepPart [80]) reported in MBS [27]. Their experimental details can be found in MBS [27]. Lastly, we train our object segmentation network using single point clouds with full annotations, denoted as OGC$_{sup}$. All implementation details are in Appendix A.4.

**Analysis:** As shown in Table 1, our OGC surpasses the classical clustering based and motion segmentation methods by large margins on all metrics, showing the advantage of our method in fully leveraging both the motion patterns and various types of geometry consistency. Compared with the fully supervised baselines, our method is only inferior to the strong MBS [27] and OGC$_{sup}$. However, we observe that our OGC actually shows a higher precision score than OGC$_{sup}$, primarily because our method tends to learn better objectness thanks to a combination of motion pattern and smoothness constraints and avoid dividing a single object into pieces. Figure 3 shows qualitative results.

### 4.2 Evaluation on OGC-DR / OGC-DRSV Datasets

We further evaluate our method to segment objects in indoor 3D scenes. Considering that the existing dataset FlyingThings3D [47] tends to have unrealistically cluttered scenes with severely fragmented objects and it is originally introduced for scene flow estimation, we turn to synthesize a new dynamic room dataset, called **OGC-DR**, that suits both scene flow estimation and object segmentation. In particular, we follow [54] to randomly place $4 \sim 8$ objects belonging to 7 classes of ShapeNet [7] {chair, table, lamp, sofa, cabinet, bench, display} into each room. In total, we create 3750, 250, and 1000 indoor rooms (scenes) for training/validation/test splits. In each scene, we create rigid dynamics by applying continuous random transformations to each object and record 4 sequential frames for evaluation. Each point cloud frame is downsampled to 2048 points. Note that, we follow [13] to split different object instances for train/val/test sets.

Table 2: Quantitative results of our method and baselines on our OGC-DR/OGC-DRSV dataset.

| | | AP↑ | PQ↑ | F1↑ | Pre↑ | Rec↑ | mIoU↑ | RI↑ |
|---|---|---|---|---|---|---|---|---|
| Supervised Method | $OGC_{sup}$ | 90.7 / 86.3 | 82.6 / 78.8 | 87.6 / 85.0 | 83.7 / 82.2 | 92.0 / 88.0 | 89.2 / 83.9 | 97.7 / 97.1 |
| Unsupervised Motion Segmentation | TrajAffn [52] | 42.6 / 39.3 | 46.7 / 43.8 | 57.8 / 54.8 | 69.6 / 63.0 | 49.4 / 48.4 | 46.8 / 45.9 | 80.1 / 77.7 |
| | SSC [51] | 74.5 / 70.3 | 79.2 / 75.4 | 84.2 / 81.5 | 92.5 / 89.6 | 77.3 / 74.7 | 74.6 / 70.8 | 91.5 / 91.3 |
| Unsupervised Methods | WardLinkage [30] | 72.3 / 69.8 | 74.0 / 71.6 | 82.5 / 80.5 | 93.9 / 91.8 | 73.6 / 71.7 | 69.9 / 67.2 | 94.3 / 93.3 |
| | DBSCAN [17] | 73.9 / 71.9 | 76.0 / 76.3 | 81.6 / 81.8 | 85.8 / 79.1 | 77.8 / 84.8 | 74.7 / 80.1 | 91.5 / 93.5 |
| | **OGC(Ours)** | **92.3 / 86.8** | **85.1 / 77.0** | **89.4 / 83.9** | 85.6 / 77.7 | **93.6 / 91.2** | **90.8 / 84.8** | **97.8 / 95.4** |

Based on our OGC-DR dataset, we collect single depth scans every time step on the mesh models to generate another dataset, called Single-View OGC-DR (**OGC-DRSV**). All object point clouds in OGC-DRSV are severely incomplete due to self- and/or mutual occlusions, resulting in the new dataset significantly more challenging than OGC-DR. Each point cloud frame in OGC-DRSV is also downsampled to 2048 points. More details of these two datasets are in Appendix A.3.

**Analysis:** As shown in Table 2, our method outperforms all classical unsupervised methods including the clustering based and the motion segmentation based methods on OGC-DR. Since the synthetic rooms in OGC-DR all have complete 3D objects, and the generated point cloud sequences are of high quality. Therefore, our OGC even surpasses the supervised $OGC_{sup}$. This shows that the rigid dynamic motions can indeed provide sufficient supervision signals to identify objects. On OGC-DRSV, our method still achieves superior performance and demonstrates robustness to incomplete point clouds, although the scores are slightly lower than that on the full point cloud dataset OGC-DR (AP: 86.8 $vs$ 92.3). Figure 3 shows qualitative results.

### 4.3 Evaluation on KITTI Scene Flow Dataset

We additionally evaluate our method on the challenging real-world outdoor KITTI Scene Flow (KITTI-SF) dataset. Officially, KITTI-SF dataset [48] consists of 200 (training) pairs of point clouds from real-world traffic scenes and an online hidden test for scene flow estimation. In our experiment, we train our pipeline on the first 100 pairs of point clouds, and then test on the remaining 100 pairs (200 single point clouds). We observe that in the 100 training pairs, the moving objects are only cars and trucks. Therefore, in the testing phase, we only keep the human annotations [1] of cars and trucks in every single frame to compute the scores. All other objects are treated as part of background. Note that, the whole background is not ignored, but counted as one object in our evaluation, and the cars and trucks can be static or moving. We find KITTI-SF is too challeging for the classical unsupervised methods, due to the extreme imbalance of 3D points between objects and background. Besides, the background and objects in KITTI-SF are always connected because of the Earth's gravity, while clustering-based WardLinkage and DBSCAN favor spatially separated objects. Therefore, we leverage the prior about ground planes in KITTI-SF to assist these methods. We detect and specially handle the ground planes, leaving above-ground points only for these methods to handle. Implementation details are in Appendix A.4.

**Analysis:** As shown in Table 3, our method obtains superior segmentation scores on the KITTI-SF dataset, being very close to our fully-supervised counterpart $OGC_{sup}$. This demonstrates the excellence of our method on real-world scenes. Figure 3 shows qualitative results.

Table 3: Quantitative results of our method and baselines on the KITTI-SF dataset.

| | | AP↑ | PQ↑ | F1↑ | Pre↑ | Rec↑ | mIoU↑ | RI↑ |
|---|---|---|---|---|---|---|---|---|
| Supervised Method | $OGC_{sup}$ | 62.4 | 52.7 | 65.1 | 63.4 | 67.0 | 67.3 | 95.0 |
| Unsupervised Motion Segmentation | TrajAffn [52] | 24.0 | 30.2 | 43.2 | 37.6 | 50.8 | 48.1 | 58.5 |
| | SSC [51] | 12.5 | 20.4 | 28.4 | 22.8 | 37.6 | 41.5 | 48.9 |
| Unsupervised Methods | WardLinkage [30] | 25.0 | 16.3 | 22.9 | 13.7 | **69.8** | 60.5 | 44.9 |
| | DBSCAN [17] | 13.4 | 22.8 | 32.6 | 26.7 | 42.0 | 42.6 | 55.3 |
| | **OGC(Ours)** | **54.4** | **42.4** | **52.4** | **47.3** | 58.8 | **63.7** | **93.6** |

### 4.4 Generalization to KITTI Detection and SemanticKITTI Datasets

Given our well trained model on KITTI-SF in Section 4.3, we directly test it on the popular KITTI 3D Object Detection (KITTI-Det) [19] and SemanticKITTI [5] benchmarks. Unlike the stereo-based

point clouds in KITTI-SF, point clouds in these two datasets are collected by LiDAR sensors and thus more sparse.

- **KITTI-Det** officially has 3712 point cloud frames for training, 3769 for validation. We only keep the ground truth object masks obtained from bounding boxes for the car category in each frame. All other objects are treated as part of background. For comparison, we download the official pretrained models of three fully-supervised methods PointRCNN [59], PV-RCNN [58] and Voxel-RCNN [14] to directly test on the validation split using the same settings as ours. In addition, we use the well trained $OGC_{sup}$ on KITTI-SF to directly test for comparison, denoted as $OGC*_{sup}$. We also train $OGC_{sup}$ on the training split (3712 frames) from scratch and test it on the remaining 3769 frames using the same evaluation settings, denoted as $OGC_{sup}$.
- **SemanticKITTI** officially has 11 sequences with annotations for training and another 11 sequences for online hidden test. We only keep ground truth objects of car and truck categories. The total 11 training sequences (23201 point cloud frames) are used for testing. Compared with KITTI-Det, SemanticKITTI holds $6\times$ more testing frames and covers more diverse scenes. Following the official split in [5], we also report the results on: i) sequences 00~07 and 09~10 (19130 frames), and ii) the sequence 08 (4071 frames) separately.

Table 4: Quantitative results on KITTI-Det (* denotes the model trained on KITTI-SF).

|  |  | AP↑ | PQ↑ | F1↑ | Pre↑ | Rec↑ | mIoU↑ | RI↑ |
|---|---|---|---|---|---|---|---|---|
| Supervised Methods | PointRCNN [59] | 95.7 | 80.1 | 88.9 | 81.3 | 98.0 | 91.4 | 97.2 |
|  | PV-RCNN [58] | 95.4 | 77.3 | 84.4 | 73.7 | 98.8 | 92.7 | 97.1 |
|  | Voxel-RCNN [14] | 95.8 | 79.6 | 87.3 | 78.1 | 98.9 | 92.6 | 97.3 |
|  | $OGC_{sup}$ | 80.0 | 68.5 | 78.3 | 72.7 | 84.8 | 84.0 | 96.9 |
|  | $OGC*_{sup}$ | 51.4 | 41.0 | 49.1 | 43.7 | 56.0 | 66.2 | 91.0 |
| Unsupervised Method | **OGC*(Ours)** | 40.5 | 30.9 | 37.0 | 30.8 | 46.5 | 60.6 | 86.4 |

**Analysis:** As shown in Tables 4&5, our method can directly generalize to 3D object segmentation on sparse LiDAR point clouds with satisfactory results, also being close to the fully-supervised counterpart $OGC*_{sup}$. It is understandable that the other three fully-supervised models have a clear advantage over ours on KITTI-Det, because they are fully supervised and trained on the KITTI-Det training split (3712 frames) while ours does not. We hope that our method can serve the first baseline and inspire more advanced unsupervised methods in the future to close the gap. Figure 3 shows qualitative results.

Table 5: Quantitative results on SemanticKITTI (* denotes the model trained on KITTI-SF).

| Sequences | Methods | AP↑ | PQ↑ | F1↑ | Pre↑ | Rec↑ | mIoU↑ | RI↑ |
|---|---|---|---|---|---|---|---|---|
| 00~10 | $OGC*_{sup}$ | 53.8 | 41.3 | 48.1 | 40.1 | 60.0 | 68.3 | 90.0 |
|  | **OGC*(Ours)** | 42.6 | 30.2 | 35.3 | 28.2 | 47.3 | 60.3 | 86.0 |
| 00~07 & 09~10 | $OGC*_{sup}$ | 55.3 | 41.8 | 48.4 | 40.1 | 61.1 | 69.9 | 90.3 |
|  | **OGC*(Ours)** | 43.6 | 30.5 | 35.5 | 28.1 | 48.2 | 62.1 | 86.3 |
| 08 | $OGC*_{sup}$ | 49.4 | 39.2 | 46.6 | 40.0 | 55.8 | 60.3 | 88.3 |
|  | **OGC*(Ours)** | 38.6 | 29.1 | 34.7 | 28.6 | 44.0 | 51.8 | 84.3 |

## 4.5 Ablation Study

**(1) Geometry Consistency Losses:** To validate the choice of our design, we firstly conduct three groups of ablative experiments on the SAPIEN dataset [75]: 1) only remove the dynamic rigid loss $\ell_{dynamic}$, 2) only remove the smoothness loss $\ell_{smooth}$, and 3) only remove the invariance loss $\ell_{invariant}$. As shown

Table 6: Ablation studies about loss designs on SAPIEN.

|  | AP↑ | PQ↑ | F1↑ | Pre↑ | Rec↑ | mIoU↑ | RI↑ |
|---|---|---|---|---|---|---|---|
| w/o $\ell_{dynamic}$ | 35.4 | 35.3 | 54.1 | **91.1** | 38.5 | 28.6 | 52.7 |
| w/o $\ell_{smooth}$ | 21.8 | 18.5 | 26.9 | 19.1 | 45.4 | 52.4 | 63.7 |
| w/o $\ell_{invariant}$ | 48.9 | 46.1 | 61.3 | 61.9 | 60.7 | 57.9 | 70.3 |
| Full OGC | **55.6** | **50.6** | **65.1** | 65.0 | **65.2** | **60.9** | **73.4** |

in Table 6, combining the proposed three losses together gives the highest segmentation scores. Basically, the dynamic rigid loss serves to discriminate multiple objects from different motion patterns. Without it, the network tends to assign all points to a single object as the shortcut to minimize the other two losses. However, we observe in Table 6 that without $\ell_{dynamic}$, the network still works to some extend. This is because the synthetic SAPIEN dataset tends to have a number of point cloud frames with only 2 or 3 objects, thus assigning all points to a single object can still get plausible

scores. This issue is further validated by conducting additional ablation experiments on curated SAPIEN dataset. More details are in Appendix A.5.

In addition, we evaluate the robustness of our object segmentation method with regard to different types of motion estimations, and different hyperparameter and design choices of our smoothness loss $\ell_{smooth}$. More results are in Appendix A.5.

**(2) Iterative Optimization Algorithm:** We also conduct ablative experiments to validate the effectiveness of our proposed Algorithm 1. We set the number of iterative rounds $R$ as $\{1, 2, 3\}$. As shown in Table 7, after 2 rounds, satisfactory segmentation results can be achieved, although we expect better results after

Table 7: Iterative optimization on SAPIEN.

| | Object Segmentation | | | | | | |
|---|---|---|---|---|---|---|---|
| #R | AP↑ | PQ↑ | F1↑ | Pre↑ | Rec↑ | mIoU↑ | RI↑ |
| 1 | 45.9 | 47.7 | 62.3 | 60.2 | 64.5 | 60.2 | 72.3 |
| 2 | 55.6 | 50.6 | 65.1 | 65.0 | 65.2 | 60.9 | 73.4 |
| 3 | 56.3 | 50.7 | 65.4 | 65.1 | 65.8 | 61.1 | 73.7 |

more rounds with longer training time. This shows that our iterative optimization algorithm can indeed fully leverage the mutual benefits between object segmentation and motion estimation.

### 4.6 Pushing the Boundaries of Unsupervised Scene Flow Estimation and Segmentation

In addition to the improvement of object segmentation from our iterative optimization algorithm, the scene flow estimation can be naturally further improved from our estimated object masks as well. Given our well trained model on the KITTI-SF dataset in Section 4.3, we use the estimated object masks to further im-

Table 8: Scene flow estimation on the KITTI-SF dataset.

| | EPE3D↓ | AccS↑ | AccR↑ | Outlier↓ |
|---|---|---|---|---|
| Ego-motion [66] | 41.54 | 22.09 | 37.21 | 80.96 |
| PointPWC-Net [74] | 25.49 | 23.79 | 49.57 | 68.63 |
| FlowStep3D [34] | 10.21 | 70.80 | 83.94 | 24.56 |
| **OGC(Ours)** | **6.72** | **80.16** | **89.08** | **22.56** |

prove the scene flow estimation. As shown in Table 8, following the exact evaluation settings of FlowStep3D [34], our method, not surprisingly, significantly boosts the scene flow accuracy, surpassing the state-of-the-art unsupervised FlowStep3D [34] and other baselines in all metrics.

In fact, our object segmentation backbone is also flexible to take the scene flow as input instead of point $xyz$ to segment objects. This is commonly called motion segmentation. We replace the single point clouds by (estimated) scene

Table 9: Motion $vs$ Points based segmentation on KITTI-SF.

| Input | AP↑ | PQ↑ | F1↑ | Pre↑ | Rec↑ | mIoU↑ | RI↑ |
|---|---|---|---|---|---|---|---|
| scene flow | 47.3 | 41.2 | 50.2 | **50.9** | 49.6 | 56.0 | 89.3 |
| point cloud | **54.4** | **42.4** | **52.4** | 47.3 | **58.8** | **63.7** | **93.6** |

flow vectors as our network inputs, and train the network from scratch using the same settings on the KITTI-SF dataset. As shown in Table 9, we can see that our network can still achieve superior results regardless of the modality of inputs, demonstrating the generality of our framework.

## 5 Conclusion

In this paper, we demonstrate for the first time that 3D objects can be accurately segmented using an unsupervised method from raw point clouds. Unlike the existing approaches which usually rely on a large amount of human annotations of every 3D object for training networks, we instead turn to leverage the diverse motion patterns over sequential point clouds as supervision signals to automatically discover the objectness from single point clouds. A series of loss functions are designed to preserve the object geometry consistency over spatial and temporal scales. Extensive experiments over multiple datasets including the extremely challenging outdoor scenes demonstrate the effectiveness of our method.

**Broader Impact:** The proposed OGC learns 3D objects from raw point clouds without requiring human annotations for supervision. We showcase the effectiveness for some basic applications including object part instance segmentation, indoor object segmentation and outdoor vehicle identification. We also believe that our method can be general for other domains such as AR/VR.

**Acknowledgements:** This work was partially supported by Shenzhen Science and Technology Innovation Commission (JCYJ20210324120603011).

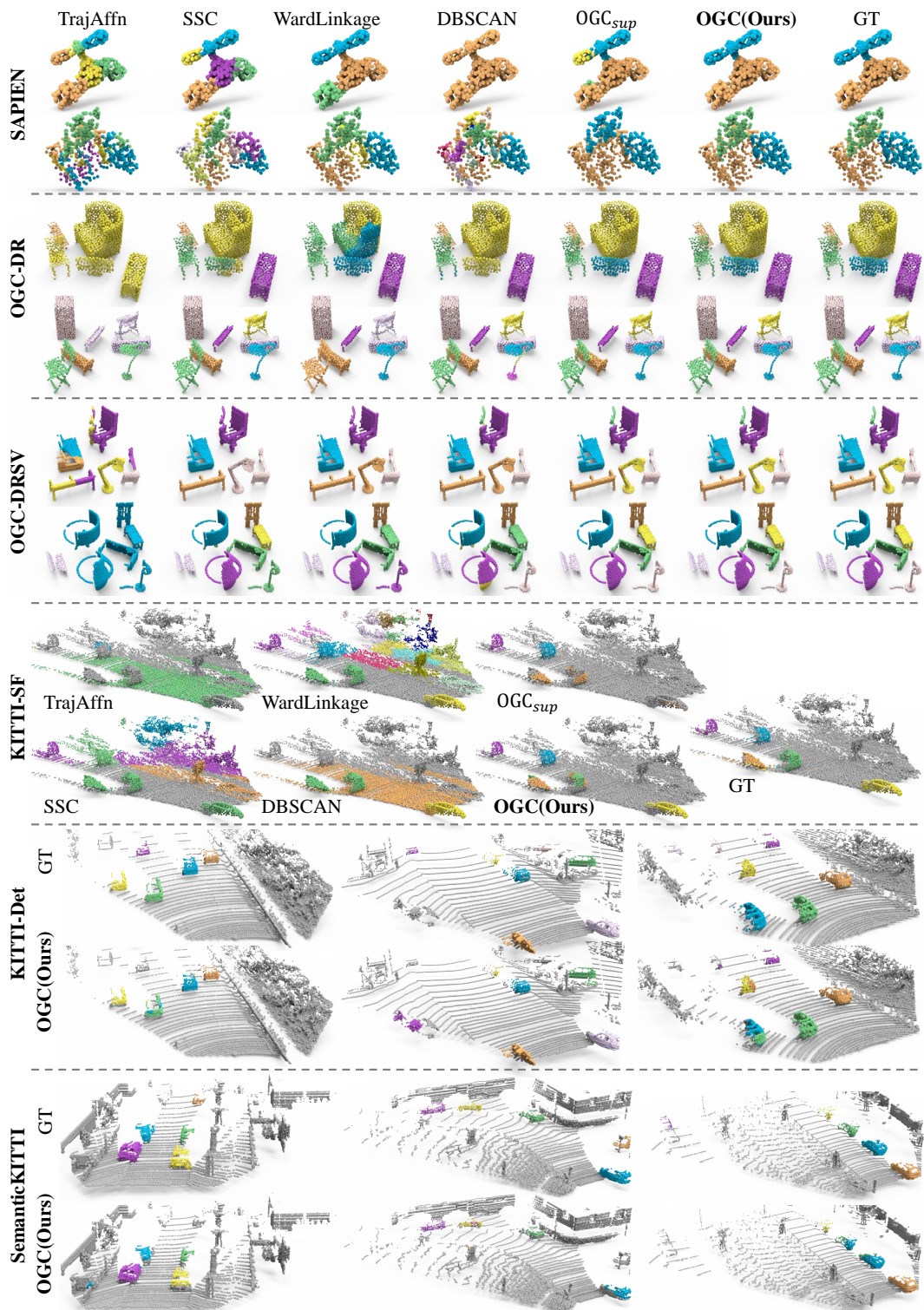

Figure 3: Qualitative results on various datasets. More qualitative results can be found in Appendix A.7 and our video demo: https://youtu.be/dZBjvKWJ4K0
.

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
