# A Appendix

## A.1 Network Architecture

We provide a detailed description of our object segmentation network and the auxiliary self-supervised scene flow estimator.

### (1) Object Segmentation Network

As shown in Figure 4, the network takes a single point cloud with $N$ points as input. It consists of Set Abstraction (SA) modules from PointNet++ [57] to extract per-point features for the downsampled point cloud with $N'$ points. Feature Propagation (FP) modules are applied subsequently to obtain per-point embeddings for all $N$ points. Given the intermediate features for the $N'$ points and the $K$ learnable queries, the standard Transformer decoder [67] is used to compute the $K$ object embeddings, each of which is expected to represent an object in the input point cloud. An MLP layer is added to reduce the dimension of object embeddings to be the same as point embeddings obtained from the PointNet++ backbone. At last, we obtain each (soft) binary mask $\boldsymbol{O}_k^t$ via a dot product between the $k^{th}$ object embedding and per-point embeddings. For each point, a softmax activation function is applied to normalize its probabilities of being assigned to different objects.

In practice, the downsampling rate and point neighborhood selection in the PointNet++ backbone are adapted to the point densities and sizes of different datasets, as shown in Table 10. The embedding dimension from the Transformer decoder is set as 128 in all datasets.

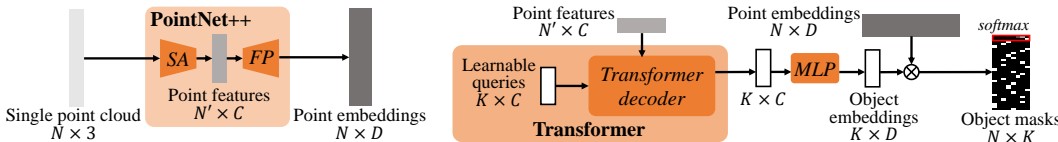

Figure 4: Detailed architecture of our object segmentation network.

Table 10: Configuration of the PointNet++ backbone in our object segmentation network. $s$ denotes the point cloud downsampling/upsampling rate. $k$ controls the $K$ nearest neighbors selected within a ball with radius $r$. $c$ denotes the first input and the following output channels of MLP layers. In SA modules, the level 1-1 and 1-2 compose a multi-scale grouping (MSG) [57] with outputs concatenated. In FP modules, the multi-level point features from SA are concatenated as inputs.

| | level | \multicolumn{4}{c}{SAPIEN / OGC-DR / OGC-DRSV} | \multicolumn{4}{c}{KITTI-SF / KITTI-Det / SemanticKITTI} |
| | | $s$ | $k$ | $r$ | $c$ | $s$ | $k$ | $r$ | $c$ |
|---|---|---|---|---|---|---|---|---|---|
| SA | 1-1 | 1/2 | 64 | 0.1(0.05) | {3,64,64} | 1/4 | 64 | 1.0 | {3,32,32,32} |
| | 1-2 | 1/2 | 64 | 0.2(0.1) | {3,64,64,128} | 1/4 | 64 | 2.0 | {3,32,32,64} |
| | 2 | 1/4 | 64 | 0.4(0.2) | {192,128,128,256} | 1/8 | 64 | 4.0 | {96,64,64,128} |
| | 3 | | | | | 1/16 | 64 | 8.0 | {128,128,128,256} |
| FP | 3 | | | | | 1/8 | | | {384,128,128} |
| | 2 | 1/2 | | | {448,256,128} | 1/4 | | | {224,64,64} |
| | 1 | 1 | | | {131,128,128,64} | 1 | | | {67,64,64,64} |

### (2) Self-Supervised Scene Flow Estimator

We use the existing FlowStep3D as our self-supervised scene flow estimator. This method extracts per-point features via a PointNet++ backbone from two input point cloud frames separately. Then it adopts a recurrent architecture to refine the scene flow predictions iteratively. We refer readers to [34] for more details. On SAPIEN and OGC-DR / OGC-DRSV datasets, with smaller scene sizes and fewer input points, we remove the last SA module with 1/32 downsampling rate and reduce the number of nearest neighbors as half of its original choice in all modules.

## A.2 Object-Aware ICP Algorithm

In Algorithm 2, we present our object-aware ICP (Iterative Closest Point) algorithm.

---

**Algorithm 2** Object-aware ICP algorithm. Assume each training sample contains a pair of point clouds and scene flow estimations $(\boldsymbol{P}^t, \boldsymbol{P}^{t+1}, \boldsymbol{M}^t \in \mathbb{R}^{N \times 3})$, and object masks $(\boldsymbol{O}^t, \boldsymbol{O}^{t+1} \in \mathbb{R}^{N \times K})$ obtained from a trained object segmentation network.

---

*Step 1: Match the individual masks in $\boldsymbol{O}^t$ and $\boldsymbol{O}^{t+1}$.*
- Use the estimated scene flows to warp the first point cloud: $\boldsymbol{P}_w^t = \boldsymbol{P}^t + \boldsymbol{M}^t$, the warped $\boldsymbol{P}_w^t$ naturally inherits the per-point object masks from $\boldsymbol{P}^t$: $\boldsymbol{O}_w^t = \boldsymbol{O}^t$.
- Compute another object masks $\hat{\boldsymbol{O}}^{t+1}$ for the second point cloud $\boldsymbol{P}^{t+1}$ using the nearest-neighbor interpolation from $(\boldsymbol{P}_w^t, \boldsymbol{O}_w^t)$.
- Leverage the Hungarian algorithm [36] to one-one match the individual masks in $\hat{\boldsymbol{O}}^{t+1}$ and $\boldsymbol{O}^{t+1}$ according to the object pair-wise IoU scores.
- Reorder the masks in $\boldsymbol{O}^{t+1}$ to align with $\hat{\boldsymbol{O}}^{t+1}$, thus aligh with $\boldsymbol{O}^t$.

*Step 2: Iteratively refine the rigid scene flow estimations*
- Compute the per-point object consistency scores $\boldsymbol{O} \in \mathbb{R}^{N \times N}$ between $\boldsymbol{P}^t$ and $\boldsymbol{P}^{t+1}$: $\boldsymbol{O} = \boldsymbol{O}^t (\boldsymbol{O}^{t+1})^\intercal$.

**for** number of iterations $I$ **do**
- Compute the per-point soft correspondence scores $\boldsymbol{C} \in \mathbb{R}^{N \times N}$ between $\boldsymbol{P}^t$ and $\boldsymbol{P}^{t+1}$ based on the nearest-neighbor (closest point) distances, where

$$\boldsymbol{C}_{ij} = exp(-\boldsymbol{\delta}_{ij}/\tau), \boldsymbol{\delta}_{ij} = \left\| \boldsymbol{p}_i^t + \boldsymbol{m}_i^t - \boldsymbol{p}_j^{t+1} \right\|_2$$

- Filter the per-point correspondence scores by the object consistency scores: $\boldsymbol{C} = \boldsymbol{C} * \boldsymbol{O}$.
- Update the scene flows $\boldsymbol{M}^t$ from the object-aware soft correspondences, where

$$\boldsymbol{m}_i^t = \frac{\sum_{j=1}^N \boldsymbol{C}_{ij}(\boldsymbol{p}_j^{t+1} - \boldsymbol{p}_i^t)}{\sum_{j=1}^N \boldsymbol{C}_{ij}}$$

- For the $k^{th}$ object, retrieve its (soft) binary mask $\boldsymbol{O}_k^t$, and then feed the tuple $\{\boldsymbol{P}^t, \boldsymbol{P}^t + \boldsymbol{M}^t, \boldsymbol{O}_k^t\}$ into weighted-Kabsch [31; 21] algorithm, estimating its transformation matrix $\boldsymbol{T}_k$.
- Update the scene flows $\boldsymbol{M}^t$ from the estimated transformations, where:

$$\boldsymbol{m}_i^t = \left( \sum_{k=1}^K o_{ik}^t \cdot (\boldsymbol{T}_k \circ \boldsymbol{p}_i^t) \right) - \boldsymbol{p}_i^t$$

Return the scene flows $\boldsymbol{M}^t$ from the last iteration.

---

In the iterative optimization, as the scene flow estimations gradually approach more consistent and accurate values, the number of iterations $I$ in the object-aware ICP can be reduced for efficiency. We set $I$ in the object-aware ICP as $\{20, 10, 5\}$ in the round $\{1, 2, 3\}$ of the iterative optimization.

Compared to the weighted-Kabsch [31; 21] algorithm, our object-aware ICP algorithm takes two frames as input to correct the inconsistency in the flows. As shown in Table 11, our algorithm obtains a larger improvement for scene flow estimations. In general, our algorithm is an extension of the classical ICP [2] to 3D scenes with multiple rigid objects. Our algorithm can be naturally implemented in a batch-wise manner, without sacrificing the optimization speed of the network.

Table 11: Scene flow estimation on KITTI-SF benchmark.

| | EPE3D↓ | AccS↑ | AccR↑ | Outlier↓ |
|---|---|---|---|---|
| FlowStep3D [34] | 10.21 | 70.80 | 83.94 | 24.56 |
| Weighted Kabsch [31; 21] | 9.31 | 71.01 | 81.20 | 28.75 |
| **Object-aware ICP** | **6.72** | **80.16** | **89.08** | **22.56** |

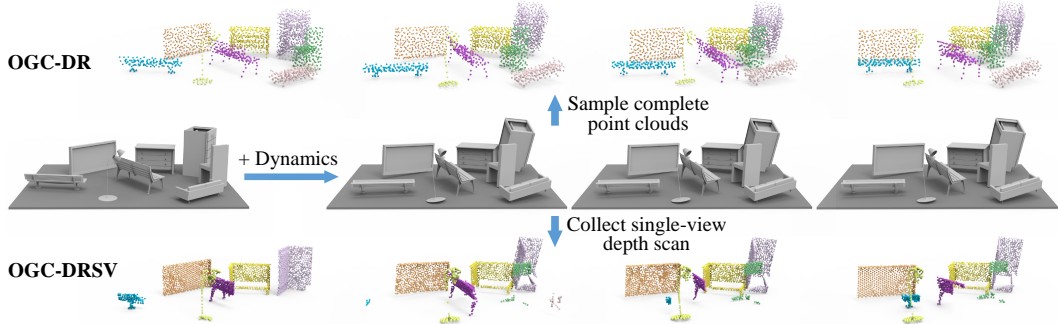

Figure 5: Illustration of the data generation process for our OGC-DR dataset.

## A.3 OGC-DR and OGC-DRSV Datasets

Here we provide details about the data generation of our OGC-DR and OGC-DRSV datasets. Following [54], we first generate 5000 static scenes with $4 \sim 8$ objects. For a single scene, the ratio of width and length of the ground plane is uniformly sampled between $0.6 \sim 1.0$. For each object in a scene, its scale is sampled from $0.2 \sim 0.45$. The object rotation angles around the vertical y-axis are randomly sampled from $-180° \sim 180°$. Unlike [54], we do not keep walls and ground planes in the generated raw point clouds since these textureless surfaces create intractable ambiguities for self-supervised scene flow estimation. In fact, it is trivial to detect and remove them via plane fitting in real-world indoor scenes. The walls and ground planes are simply used to place all objects in a more realistic manner.

We then create rigid dynamics for the objects. For each object in a scene, we sample a rigid transformation relative to its pose in the previous frame. In particular, we first uniformly sample an angle from $-10° \sim 10°$ rotated by y/x/z axis with the probability of {0.6, 0.2, 0.2} respectively. Afterwards, we uniformly sample a translation only on the x-z plane from the range of $-0.04 \sim 0.04$ for each object, ensuring that the object is always on the ground. Note that, we reject all samples which have objects overlapped or being out of the scene boundary.

The last point cloud sampling step varies for the two datasets. For OGC-DR, we directly sample point clouds from the surfaces of complete mesh models, while for OGC-DRSV, we collect single-view depth scans on mesh models. Note that for both datasets, the point sampling is independently conducted on each frame in a scene. Therefore, there is no exact point correspondences between consecutive frames. This setting is consistent with the scene flow estimation task in general real-world scenes. Figure 5 illustrates the complete data generation process.

## A.4 Additional Implementation Details

### (1) Data Preparation

**SAPIEN:** In SAPIEN, each scene (an articulated object) has 4 sequential scans. During training, we leverage consecutive frame pairs (both forward and backward) only, because the self-supervised scene flow estimator can hardly handle rapid motions. Therefore, each object has 6 pairs of point clouds. Given 13682/2356 objects in training/validation splits, we get 82092 training and 14136 validation frame pairs. The 720 objects in testing split contribute 2880 individual frames for evaluation.

**OGC-DR/OGC-DRSV:** Similar to SAPIEN, each scene in OGC-DR/OGC-DRSV holds 4 sequential frames. The 3750/250 scenes in training/validation splits give 22500 training and 1500 validation frame pairs, and the 1000 scenes in the testing split provide 4000 testing frames.

**KITTI-SF:** The 100 pairs of point clouds in the training split of KITTI-SF contribute 200 training pairs (both forward and backward), and the other 100 pairs in testing split provide 200 individual frames for evaluation. A tricky problem in KTTI-SF is the scene flow estimation for the ground. The textureless ground poses intractable ambiguities for the self-supervised scene flow estimator. However, we cannot simply remove the ground by applying a height threshold, because the background points above the ground will no longer be spatially connected, thus breaking the assumption behind our geometry smoothness regularizer $\ell_{smooth}$. To address this issue, we apply the self-supervised scene

flow estimator to points above the ground only. Meanwhile, we use the classical ICP [2] algorithm onto points above the ground and regard the fitted transformation as motions for ground points (*i.e.*, the camera ego-motion). Although this solution relies on an assumption that static background points dominate the scene, our object-aware ICP algorithm can empirically alleviate potential errors in the iterative optimization.

### (2) Hyperparameter Selection

**Geometry Smoothness Regularization:** We choose $L1$ for the distance function $d()$ given its less sensitivity to "outliers", *i.e.*, adjacent points belonging to different objects. As shown in Table 12, we select two groups of neighboring points from two different scales (denoted by $\{k_1, r_1\}$ and $\{k_2, r_2\}$) and weight them by $\{3.0, 1.0\}$ in $\ell_{smooth}$.

**Geometry Invariance Loss:** For the distance function $\hat{d}()$ here, $L1$, $L2$ and cross-entropy are all theoretically reasonable. We choose $L2$ for the best performance. The transformation for augmentation comprises a scale factor uniformly sampled from $0.95 \sim 1.05$ and a rotation around the vertical y-axis sampled from $-180 \sim 180°$. On KITTI-SF dataset, we add an x-z translation sampled from $-1 \sim 1$ and a y translation sampled from $-0.1 \sim 0.1$.

**Network Training:** We adopt the Adam optimizer with a learning rate of 0.001 and train on SAPIEN/OGC-DR/KITTI-SF datasets for 40/40/200 epochs, respectively. The batch size is set as 32/8/4 on each dataset to fill in the whole memory of a single RTX3090 GPU. The three losses $\ell_{dynamic}$, $\ell_{smooth}$ and $\ell_{invariant}$ are weighted by $\{10.0, 0.1, 0.1\}$. On SAPIEN

Table 12: The choices of neighboring points in the geometry smoothness regularization. $k$ controls the $K$ nearest neighbors selected within a ball with radius $r$.

| SAPIEN / OGC-DR | | | | KITTI-SF | | | |
|---|---|---|---|---|---|---|---|
| $k_1$ | $r_1$ | $k_2$ | $r_2$ | $k_1$ | $r_1$ | $k_2$ | $r_2$ |
| 8 | 0.1(0.02) | 16 | 0.2(0.04) | 32 | 1.0 | 64 | 2.0 |

and OGC-DR, since $\ell_{invariant}$ can slow down the convergence, we first train 20 epochs without it and then add it back. We also find that $\ell_{smooth}$ occasionally overwhelms the initial iterations of training, causing network predictions to collapse and assign all points to a single object. Therefore, we empirically disable $\ell_{smooth}$ before iterating through the initial 2000/2000/200 samples on SAPIEN/OGC-DR/KITTI-SF datasets.

### (3) Baseline methods on KITTI-SF

Since the KITTI-SF dataset is too challeging for the classical unsupervised methods, we leverage the prior about ground planes in the dataset to improve baseline methods. First, we detect and temporarily remove the ground plane, letting the baseline algorithms to segment above-ground points only. For TrajAffn and SSC, we can use motion information to merge the ground points with above-ground segments that are likely to be part of the static background. To do this, we employ the Kabsch algorithm to estimate the rigid transformation of the ground. Then the above-ground segments whose motions are well fitted by the ground's transformation will be incorporated. For WardLinkage and DBSCAN, the ground is treated as a separate segment. We conduct ablation studies to validate the use of ground plane prior for these baseline methods on the KITTI-SF dataset, as shown in Table 13 and Figure 6. After using the ground plane prior, SSC and WardLinkage gain remarkable improvements both quantitatively and qualitatively. For TrajAffn and DBSCAN, although the quantitative performance gain is not significant, we find that their qualitative results become more meaningful.

Table 13: Ablation studies about the ground plane prior for baseline methods on KITTI-SF.

| | use ground plane prior | AP↑ | PQ↑ | F1↑ | Pre↑ | Rec↑ | mIoU↑ | RI↑ |
|---|---|---|---|---|---|---|---|---|
| TrajAffn [52] | | 30.4 | 34.7 | 42.7 | 40.5 | 45.3 | 49.0 | 83.5 |
| | ✓ | 24.0 | 30.2 | 43.2 | 37.6 | 50.8 | 48.1 | 58.5 |
| SSC [51] | | 2.9 | 5.2 | 7.5 | 6.2 | 9.5 | 19.3 | 33.2 |
| | ✓ | 12.5 | 20.4 | 28.4 | 22.8 | 37.6 | 41.5 | 48.9 |
| WardLinkage [30] | | 1.3 | 2.4 | 3.8 | 2.2 | 14.3 | 26.8 | 15.7 |
| | ✓ | 25.0 | 16.3 | 22.9 | 13.7 | 69.8 | 60.5 | 44.9 |
| DBSCAN [17] | | 14.8 | 29.9 | 32.9 | 46.5 | 25.4 | 31.3 | 84.8 |
| | ✓ | 13.4 | 22.8 | 32.6 | 26.7 | 42.0 | 42.6 | 55.3 |

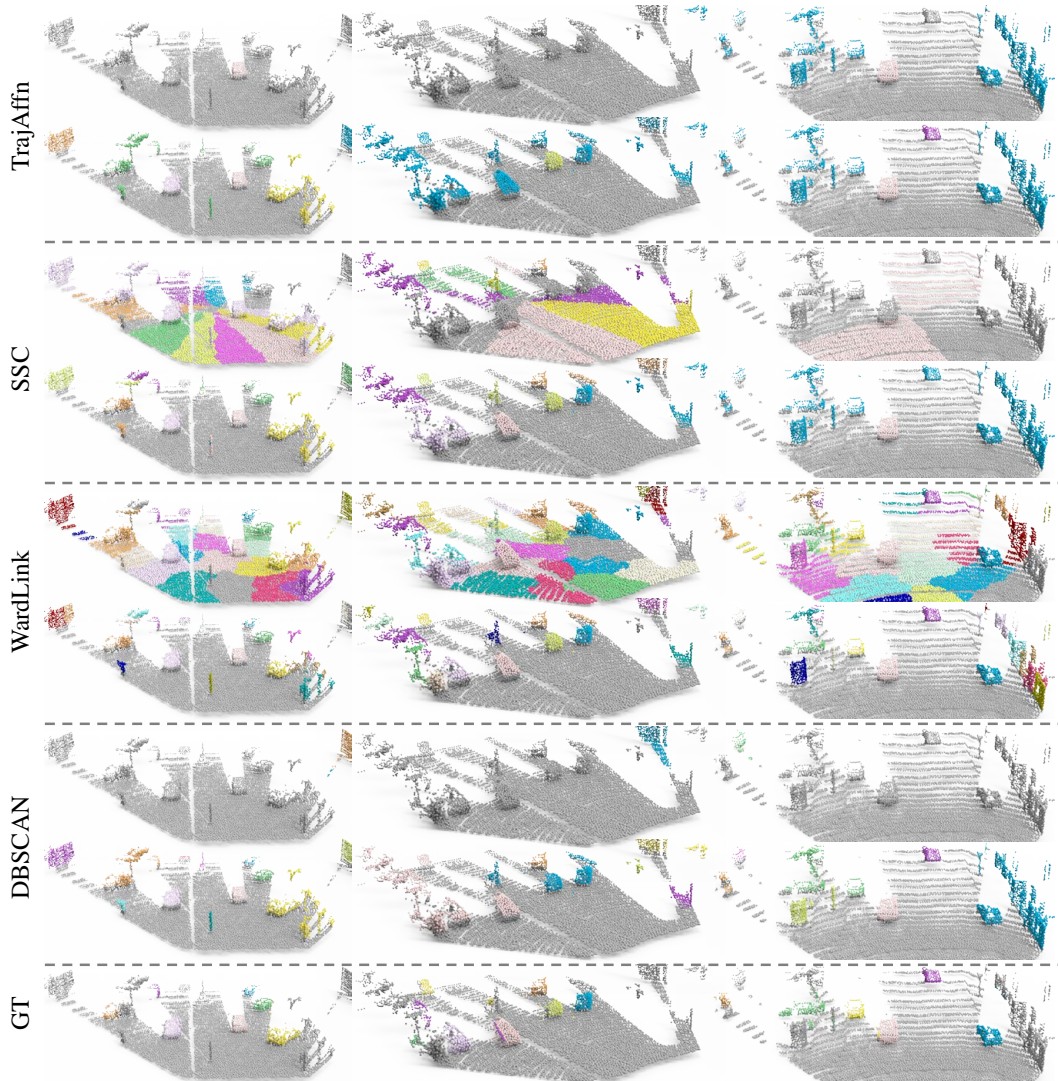

Figure 6: Qualitative results for ablation studies about the ground plane prior on KITTI-SF. For each baseline method, segmentation results without (top row) and with (bottom row) the ground plane prior are shown.

## A.5 Additional Ablation Studies

### (1) Geometry Consistency Losses

We conduct additional ablation experiments on the curated SAPIEN dataset for a more comprehensive analysis of losses in our framework. Recall that without the dynamic rigid loss $\ell_{dynamic}$, network predictions collapse and assign all points to a single object. The full SAPIEN dataset holds a number of point cloud frames with only 2 or 3 object parts, enabling the ablated model without $\ell_{dynamic}$ to still get plausible scores. However, as shown in Tables 14&15, once we evaluate the ablated models on point clouds with $\geq 3$ object parts (Table 14) or $\geq 4$ object parts (Table 15), the performance of the ablated model without $\ell_{dynamic}$ drops rapidly. This clearly shows that the $\ell_{dynamic}$ loss is truly critical to tackle complex scenes with more and more objects. Figure 7 shows qualitative examples.

We also conduct additional ablative experiments on the KITTI-SF dataset, as shown in Table 16. Similar to the results on the SAPIEN dataset, we observe the collapse of object segmentation without $\ell_{dynamic}$ and the oversegmentation issue without $\ell_{smooth}$. Figure 7 gives an intuitive illustration.

Table 14: Additional ablation results on a curated SAPIEN dataset where only point clouds with $\geq 3$ object parts are kept (844 frames in total).

| Config. | AP↑ | PQ↑ | F1↑ | Pre↑ | Rec↑ | mIoU↑ | RI↑ |
|---|---|---|---|---|---|---|---|
| w/o $\ell_{dynamic}$ | 15.8 | 21.0 | 32.7 | **69.7** | 21.4 | 18.4 | 44.1 |
| w/o $\ell_{smooth}$ | 13.0 | 15.5 | 23.7 | 18.9 | 31.8 | 42.9 | 65.5 |
| w/o $\ell_{invariant}$ | 23.8 | 28.3 | 41.3 | 47.6 | 36.5 | 38.2 | 64.0 |
| Full OGC | **30.8** | **34.0** | **48.2** | 52.4 | **44.6** | **43.4** | **67.4** |

Table 15: Additional ablation results on a curated SAPIEN dataset where only point clouds with $\geq 4$ object parts are kept (120 frames in total).

| Config. | AP↑ | PQ↑ | F1↑ | Pre↑ | Rec↑ | mIoU↑ | RI↑ |
|---|---|---|---|---|---|---|---|
| w/o $\ell_{dynamic}$ | 10.8 | 12.9 | 22.4 | **65.0** | 13.5 | 11.1 | 35.0 |
| w/o $\ell_{smooth}$ | 12.8 | 13.9 | 22.2 | 20.3 | 24.5 | **35.3** | **67.3** |
| w/o $\ell_{invariant}$ | 15.7 | 21.6 | 31.9 | 46.2 | 24.3 | 26.8 | 60.3 |
| Full OGC | **22.3** | **26.6** | **40.1** | 55.0 | **31.6** | 29.7 | 59.8 |

Table 16: Additional ablation studies about loss designs on KITTI-SF.

| Config. | AP↑ | PQ↑ | F1↑ | Pre↑ | Rec↑ | mIoU↑ | RI↑ |
|---|---|---|---|---|---|---|---|
| w/o $\ell_{dynamic}$ | 24.8 | 37.1 | 39.6 | **100.0** | 24.7 | 31.3 | 88.3 |
| w/o $\ell_{smooth}$ | 44.9 | 31.8 | 39.6 | 30.9 | 55.1 | 61.2 | 90.2 |
| w/o $\ell_{invariant}$ | 47.1 | 35.0 | 43.0 | 35.5 | 54.4 | 60.7 | 92.5 |
| Full OGC | **54.4** | **42.4** | **52.4** | 47.3 | **58.8** | **63.7** | **93.6** |

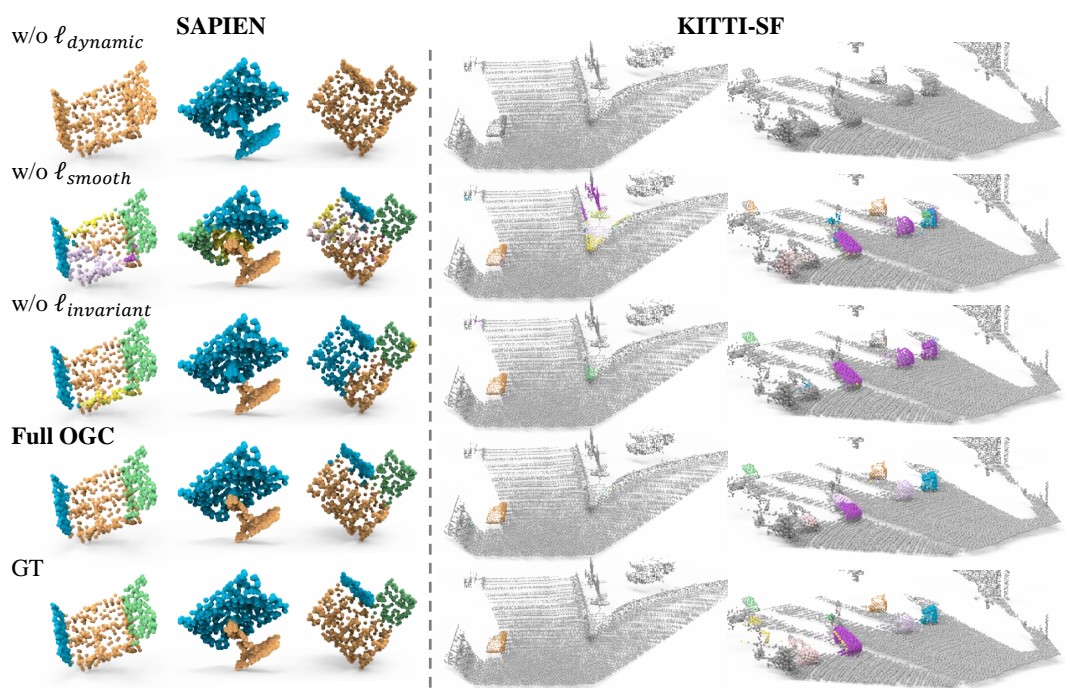

Figure 7: Qualitative results for ablation study on SAPIEN and KITTI-SF.

### (2) Use of the Invariance Loss in Iterative Optimization

We conduct ablation experiments on the KITTI-SF dataset to validate our choice of using the invariance loss $l_{invariant}$ only in the final round of iterative optimization. To do this, we iteratively optimize on KITTI-SF for 2 rounds, with two different configurations: (i) We use $l_{invariant}$ in object segmentation optimization of the two rounds. (ii) We use $l_{invariant}$ only in the 2nd round. For each configuration, we run for five times with different random seeds and report the mean results with uncertainty levels.

Table 17: Ablation results about the use of the invariance loss in iterative optimization. $\#R$ denotes the number of iterative optimization rounds.

| $\#R$ | Split | Config | AP↑ | PQ↑ | F1↑ | Pre↑ | Rec↑ | mIoU↑ | RI↑ |
|---|---|---|---|---|---|---|---|---|---|
| 1 | Train | (i) | 40.7±2.1 | 27.7±0.8 | 38.7±1.1 | 27.0±0.9 | 68.0±1.6 | 61.5±0.8 | 62.9±1.1 |
| | | (ii) | 41.4±3.0 | 31.3±0.9 | 43.5±1.2 | 30.5±1.1 | 75.4±1.2 | 64.8±0.7 | 60.6±1.0 |
| | Test | (i) | 34.1±2.0 | 24.1±1.6 | 35.0±2.1 | 26.0±1.8 | 53.9±2.2 | 52.9±1.1 | 57.2±0.9 |
| | | (ii) | 29.1±1.9 | 22.7±1.0 | 33.7±1.6 | 24.9±1.4 | 52.1±1.6 | 51.2±0.8 | 54.2±1.1 |
| 2 | Train | (i) | 57.0±8.1 | 41.3±7.3 | 52.4±7.2 | 41.9±9.2 | 71.7±2.1 | 69.2±3.6 | 83.1±11.1 |
| | | (ii) | 67.0±2.3 | 50.5±2.3 | 61.5±2.4 | 53.1±3.6 | 73.1±1.4 | 73.7±0.9 | 95.5±0.3 |
| | Test | (i) | 41.9±8.2 | 33.0±6.4 | 43.0±6.1 | 35.8±7.6 | 54.7±3.0 | 58.4±4.0 | 79.4±13.0 |
| | | (ii) | 51.8±2.2 | 40.8±2.2 | 50.6±2.6 | 45.5±3.5 | 57.2±1.7 | 62.1±1.2 | 93.4±0.3 |

**Analysis:** As shown in Table 17, in the 1st round, the model (ii) trained without $l_{invariant}$ sacrifices some generalization performance on testing data (F1: 33.7 $vs$ 35.0) while fitting better on training data (F1: 43.5 $vs$ 38.7). As shown in Table 18, such advantages in segmentation performance on training data lead to more refined scene flows (EPE3D: 2.36 $vs$ 3.12), which will directly influence the optimization in the next round. In constrast, better segmentation on testing data cannot be passed to the next round. In the 2nd round, both models are trained with $l_{invariant}$. The model (ii) stably produces superior segmentation results on both training (F1: 61.5±2.4 $vs$ 52.4±7.2) and testing data (F1: 50.6±2.6 $vs$ 43.0±6.1), owing to higher quality scene flow refinement inherited from the previous round.

The findings above are consistent with our theoretical expectations. In the iterative optimization, the previous rounds can only influence the final results by passing refined scene flows (on training split) to the following rounds. The invariance loss $l_{invariant}$ brings better generalization, especially to static objects, while these properties are of little use for scene flow refinement on training data.

Table 18: Refined scene flow estimation after the 1st round on KITTI-SF dataset.

| Config | EPE3D↓ | AccS↑ | AccR↑ | Outlier↓ |
|---|---|---|---|---|
| (i) | 3.12±0.16 | 92.8±0.7 | 94.7±0.5 | 23.9±0.5 |
| (ii) | 2.36±0.12 | 94.7±0.5 | 96.4±0.2 | 22.5±0.1 |

Therefore, in the previous rounds of iterative optimization, we can exclude $l_{invariant}$ and let the model focus on moving objects in training samples and produce more refined scene flows. In the last round, $l_{invariant}$ can be included to boost the generalization ability of the final model.

### (3) Robustness to Scene Flow Distortions

We investigate the robustness of our method to scene flow distortions on the OGC-DR and KITTI-SF datasets. To do this, we conduct experiments on three types of distorted scene flows: (i) We add different degrees of zero-mean Gaussian noise into the ground truth scene flows, and these noisy scene flows are used to supervised our object segmentation network. (ii) We use insufficiently trained scene flow estimators to produce low-quality scene flow estimations for supervision. (iii) We use the initial scene flow estimations which has not been refined by our iterative optimization. On our OGC-DR dataset, we evaluate all these ablations. On the KITTI-SF dataset, we only evaluate (i) and (iii), as we use a FlowStep3D model publicly released by the authors to estimate scene flows for KITTI-SF. The intermediate training models are not available to evaluate (ii).

**Analysis on OGC-DR:** As shown in Table 19, our OGC is robust to Gaussian noises in scene flows. The model maintains 85.2 AP even when the AccR of scene flows degrades to 6.9 only. In contrast, the flow distortions from insufficiently trained estimators incur a notable drop in the segmentation performance. The AP drops to 84.7 even when the scene flow AccR is still 32.2. From this, we hypothesize that our OGC can be robust to noisy flows with large variance thanks to the rigid loss integrated with weighted-Kabsch algorithm, but sensitive to large biases in estimated scene flows.

**Analysis on KITTI-SF:** As shown in Table 20, our method has strong robustness to Gaussian noise, same as on OGC-DR dataset. The model achieves 59.5 AP even when the scene flow is corrupted by

Table 19: Ablation results about the robustness to scene flow distortions on OGC-DR. Bold text denotes **the configuration of full OGC.** #$R$ denotes the number of iterative optimization rounds. We report the object segmentation performance on the testing set and scene flow quality on training set (the scene flow quality of the testing set is irrelevant to our object segmentation).

| | Flow Source | #$R$ | Object Segmentation | | | | | | | Scene Flow | |
|---|---|---|---|---|---|---|---|---|---|---|---|
| | | | AP↑ | PQ↑ | F1↑ | Pre↑ | Rec↑ | mIoU↑ | RI↑ | EPE3D↓ | AccR↑ |
| Ablation (i) | GT + Gaussian (std=1.0) | 1 | 91.3 | 83.3 | 88.1 | 84.4 | 92.1 | 89.2 | 97.6 | 1.60 | 73.9 |
| | GT + Gaussian (std=2.0) | 1 | 86.4 | 79.7 | 86.2 | 85.0 | 87.4 | 83.8 | 96.5 | 3.19 | 19.9 |
| | GT + Gaussian (std=3.0) | 1 | 85.2 | 77.2 | 84.5 | 82.9 | 86.2 | 82.0 | 95.8 | 4.79 | 6.9 |
| Ablation (ii) | FlowStep3D (epoch=20) | 1 | 90.1 | 81.0 | 86.4 | 81.6 | 91.9 | 88.2 | 96.8 | 1.14 | 90.2 |
| | FlowStep3D (epoch=10) | 1 | 89.1 | 79.8 | 86.1 | 82.1 | 90.6 | 86.2 | 96.3 | 1.45 | 86.0 |
| | FlowStep3D (epoch=1) | 1 | 84.7 | 71.6 | 80.5 | 74.3 | 87.8 | 80.9 | 93.5 | 3.84 | 32.2 |
| Ablation (iii) | FlowStep3D (epoch=50) | 1 | 91.3 | 83.7 | 88.4 | 84.5 | 92.7 | 89.7 | 97.6 | 0.98 | 90.0 |
| | **FlowStep3D (epoch=50)** | **2** | **92.3** | **85.1** | **89.4** | **85.6** | **93.6** | **90.8** | **97.8** | **0.76** | **92.2** |

Table 20: Ablation results about the robustness to scene flow distortions on KITTI-SF. Bold text denotes **the configuration of full OGC.**

| | Flow Source | #$R$ | Object Segmentation | | | | | | | Scene Flow | |
|---|---|---|---|---|---|---|---|---|---|---|---|
| | | | AP↑ | PQ↑ | F1↑ | Pre↑ | Rec↑ | mIoU↑ | RI↑ | EPE3D↓ | AccR↑ |
| Ablation (i) | GT + Gaussian (std=10.0) | 1 | 61.1 | 49.5 | 59.5 | 54.9 | 65.0 | 68.8 | 94.5 | 15.96 | 35.8 |
| | GT + Gaussian (std=20.0) | 1 | 59.5 | 48.5 | 58.5 | 54.4 | 63.4 | 67.3 | 94.3 | 31.92 | 7.5 |
| Ablation (iii) | FlowStep3D (epoch=120) | 1 | 36.0 | 24.6 | 35.4 | 26.4 | 53.8 | 53.7 | 57.8 | 12.21 | 72.8 |
| | **FlowStep3D (epoch=120)** | **2** | **54.4** | **42.4** | **52.4** | **47.3** | **58.8** | **63.7** | **93.6** | **2.29** | **96.3** |

Gaussian noise to only 7.5 AccR. In contrast, the model without iterative optimization only gives 36.0 AP when the scene flow AccR is 72.8. Figure 8 shows qualitative results. In the middle column, although scene flows have an overall high quality, the inconsistency in scene flows between two parts of the same object leads to over-segmentation. We believe the Weighted Kabsch algorithm inside our dynamic rigid loss is the key. This algorithm inherently smooths the Gaussian-like noise in scene flows but cannot handle the biased errors. Fortunately, our object-aware ICP is designed to correct such inconsistency in the flows, thus improving segmentation performance in iterative optimization.

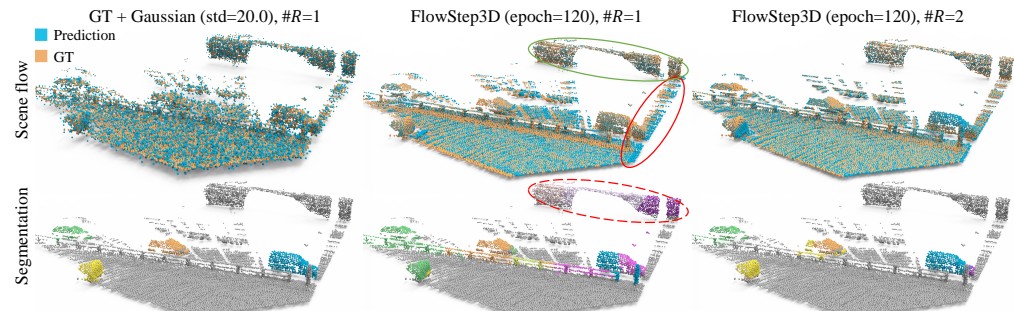

Figure 8: Qualitative results for ablation study about the robustness to scene flow distortions on KITTI-SF. Scene flows are visualized via the point cloud warped by the flows. In the middle column, the scene flow estimations are accurate for the above-ground background points (solid green ellipse) but have biased errors for the ground plane points (which can be clearly seen inside the solid red ellipse). This deviation of scene flow estimations between the above-ground background and the ground plane leads to over-segmentation of the background (dashed red ellipse).

### (4) Choice of Hyperparameters for Smoothness Regularization

We evaluate the influence of smoothness regularization hyperparameters on OGC-DR, as shown in Table 21. When we strengthen the regularization by enforcing smoothness in a larger local neighborhood (*i.e.*, ablation $H_3$), the Precision score improves with less over-segmentation, while the Recall score is sacrificed. Figure 9 shows qualitative results. In general, as expected, such hyperparameters control the trade-off between over- and under-segmentation issues.

### (5) Weighted Smoothness Regularization via Motion Similarity

We investigate a variant of the smoothness regularization which is weighted by the inter-point motion similarity. This motion-similarity-weighted smoothness regularization is mathematically defined as,

$$\ell'_{smooth} = \frac{1}{N} \sum_{n=1}^{N} \left( \frac{1}{H} \sum_{h=1}^{H} d(\boldsymbol{o}_n, \boldsymbol{o}_{n_h}) \cdot \frac{exp(-\|\boldsymbol{m}_n - \boldsymbol{m}_{n_h}\|_2 / \tau)}{E} \right) \qquad (4)$$

Table 21: Different choices of smoothness regularization hyperparamters on the OGC-DR dataset. The hyperparameter $k$ controls the $K$ nearest neighbors selected within a ball with radius $r$.

|  | $(k_1, r_1), (k_2, r_2)$ | AP↑ | PQ↑ | F1↑ | Pre↑ | Rec↑ | mIoU↑ | RI↑ |
|---|---|---|---|---|---|---|---|---|
| $H_1$ | (4, 0.01), (8, 0.02) | 92.2 | 83.4 | 88.1 | 83.1 | 93.7 | 90.5 | 97.6 |
| $H_2$ **(Full OGC)** | (8, 0.02), (16, 0.04) | 92.3 | 85.1 | 89.4 | 85.6 | 93.6 | 90.8 | 97.8 |
| $H_3$ | (32, 0.08), (64, 0.16) | 84.6 | 81.0 | 87.4 | 89.5 | 85.4 | 82.3 | 96.8 |

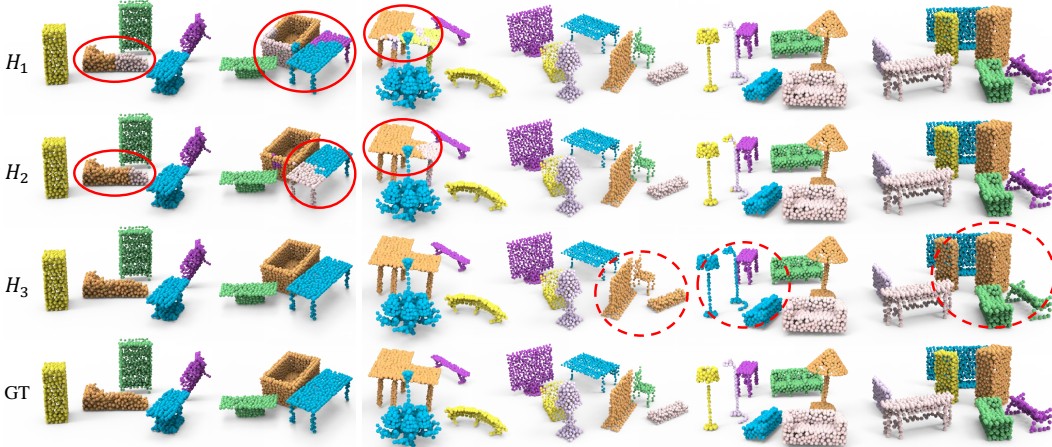

Figure 9: Qualitative results for the influence of smoothness regularization hyperparamters on OGC-DR. $H_3$ reduces over-segmentation issues in $H_1/H_2$ (solid red ellipses in column $1 \sim 3$), but fails to separate different objects sometimes (dashed red ellipses in column $4 \sim 6$).

where $\boldsymbol{m}_n \in \mathbb{R}^3$ represents the motion vector of center point $\boldsymbol{p}_n$, and $\boldsymbol{m}_{n_h} \in \mathbb{R}^3$ represents the motion vector of its $h^{th}$ neighbouring point. $\tau = 0.01$ is a temperature factor. $E$ is a normalization term, $i.e.$, $E = \sum_{h=1}^{H} exp(-\|\boldsymbol{m}_n - \boldsymbol{m}_{n_h}\|_2/\tau)$. This variant selectively enforces the object mask smoothness among points with close locations and similar motions. Intuitively, it may avoid blurry predictions on object boundaries.

Table 22: Quantitative results of motion-similarity-weighted smoothness regularization on KITI-SF. Bold text denotes **the configuration of full OGC.**

| Flow Source | #R | Regularizer | AP↑ | PQ↑ | F1↑ | Pre↑ | Rec↑ | mIoU↑ | RI↑ |
|---|---|---|---|---|---|---|---|---|---|
| **FlowStep3D (epoch=120)** | **2** | $l_{smooth}$ | 54.4 | 42.4 | 52.4 | 47.3 | 58.8 | 63.7 | 93.6 |
| FlowStep3D (epoch=120) | 2 | $l'_{smooth}$ | 49.8 | 40.7 | 50.1 | 46.0 | 55.0 | 61.1 | 93.5 |
| GT + Gaussian (std=10.0) | 1 | $l_{smooth}$ | 61.1 | 49.5 | 59.5 | 54.9 | 65.0 | 68.8 | 94.5 |
| GT + Gaussian (std=10.0) | 1 | $l'_{smooth}$ | 60.0 | 48.1 | 58.5 | 54.0 | 63.9 | 66.9 | 94.5 |
| GT + Gaussian (std=20.0) | 1 | $l_{smooth}$ | 59.5 | 48.5 | 58.5 | 54.4 | 63.4 | 67.3 | 94.3 |
| GT + Gaussian (std=20.0) | 1 | $l'_{smooth}$ | 57.9 | 46.5 | 56.3 | 51.1 | 62.6 | 67.2 | 94.4 |

**Analysis:** As shown in Table 22, $l'_{smooth}$ brings no benefits to our OGC method under various scene flow situations. We believe the weighting via motion similarity makes $l'_{smooth}$ more sensitive to noises in scene flow estimations, thus being inferior to our $l_{smooth}$.

## A.6   Limitations of OGC

Our method can neither segment non-rigid objects nor discover unseen object types due to the lack of supervision signals. Besides, the trained OGC model may not provide generalizable intermediate representations to boost the performance of supervised model, which is discussed in details below.

We investigate whether our unsupervised method OGC can be used as a pre-training technique before fully-supervised fine-tuning with a small amount of labeled data, like other popular self-supervised representation learning methods. To do this, we firstly keep a subset (10%) of labelled point clouds

Table 23: Quantitative results of OGC as a pre-training step on KITTI-Det.

| training strategy | AP↑ | PQ↑ | F1↑ | Pre↑ | Rec↑ | mIoU↑ | RI↑ |
|---|---|---|---|---|---|---|---|
| train OGC$_{sup}$ on 10% labelled KITTI-Det | 71.5 | 58.3 | 68.4 | 61.8 | 76.7 | 79.1 | 96.0 |
| (train OGC on unlabelled KITTI-SF + finetune on 10% labelled KITTI-Det) | 66.4 | 53.6 | 63.3 | 56.0 | 72.7 | 76.2 | 95.1 |
| train OGC on unlabeled KITTI-SF | 41.0 | 30.9 | 37.7 | 31.4 | 47.0 | 59.9 | 85.0 |

from KITTI-Det training set, and then use our OGC model unsupervisedly trained on KITTI-SF to be fine-tuned on the labelled subset. For comparison, we additionally train a new model with full supervision on the same labelled subset from scratch.

**Analysis:** From Table 23, we can see that: 1) Not surprisingly, using the pre-trained OGC model followed by fine-tuning brings a significant improvement over our unsupervised OGC model, *i.e.*, the 1st row *vs* the 3rd row. 2) However, the (pre-train + fine-tune) strategy fails to outperform the fully-supervised model from scratch, *i.e.*, the 1st row *vs* the 2nd row. Fundamentally, this is because our unsupervised OGC is not dedicated to learn general intermediate representations for multiple downstream tasks. Instead, our OGC is task-driven and it aims to directly segment objects from raw point clouds. The learned latent representations in unsupervised training are likely to be different from the latent representations learned in fully-supervised training. In this regard, a naïve combination of (pre-train + fine-tune) may confuse the network and give inferior results. Nevertheless, how to effectively leverage the unsupervised model along with full supervision is an interesting direction and we leave it for future exploration.

## A.7   Additional Qualitative Results

We provide additional qualitative results in Figure 11 for experiments in Sections 4.1 and 4.2 on SAPIEN and OGC-DR datasets, and in Figure 12 for experiments in Section 4.3 and 4.4 on the KITTI datasets.

For better visualization, we additionally project the segmented point clouds onto the corresponding RGB images in KITTI-SF and KITTI-Det datasets. As shown in Figure 10, our method OGC can successfully segment static cars parking alongside the road, thanks to our geometry invariance loss $l_{invariant}$ which enables our network to generalize the segmentation strategy to similar yet static objects through a set of scene transformations.

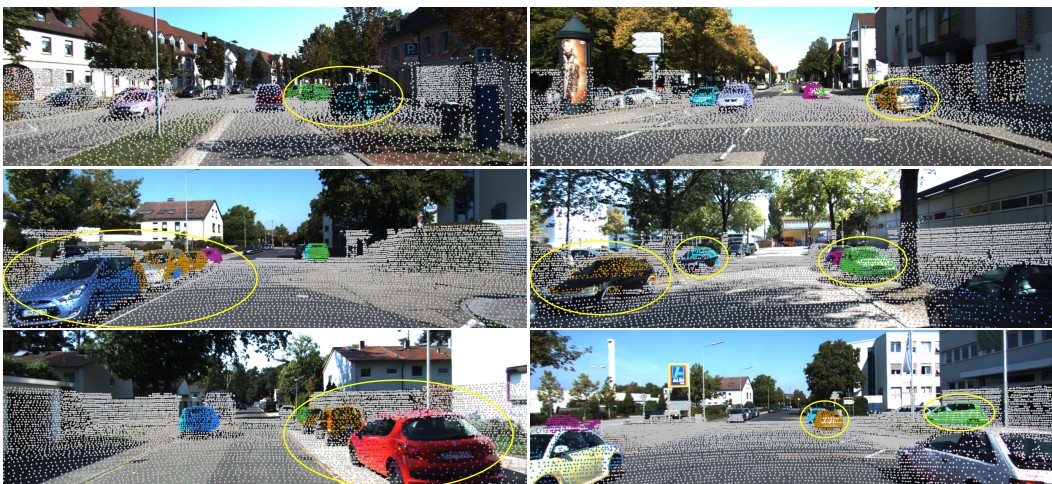

Figure 10: Qualitative results for static object segmentation on KITTI. Images in the 1st row are from KITTI-SF dataset, the rest are from KITTI-Det dataset. Static cars in yellow ellipses are parking alongside the road and can be successfully segmented by our method.

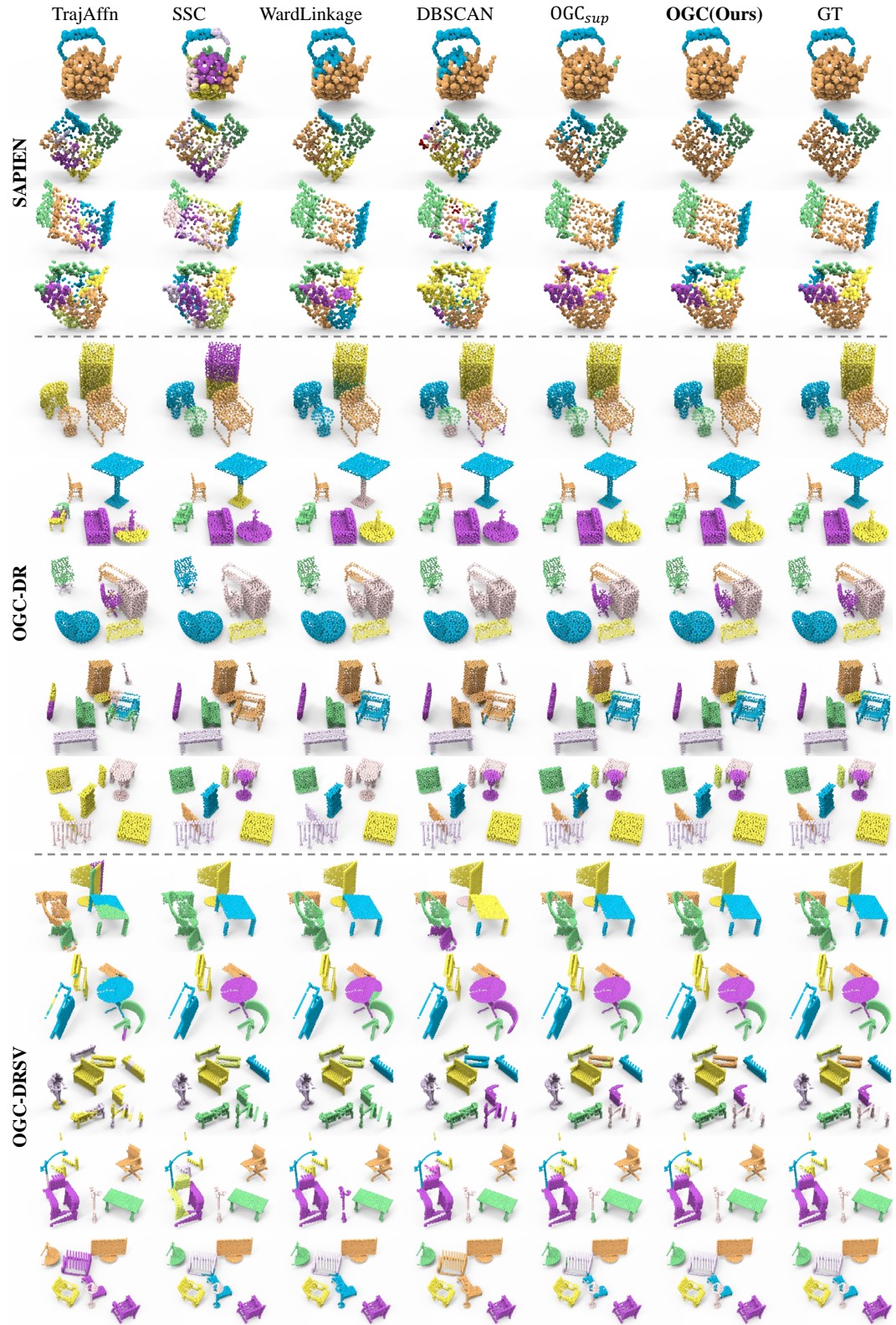

Figure 11: Additional qualitative results on SAPIEN, OGC-DR, and OGC-DRSV.

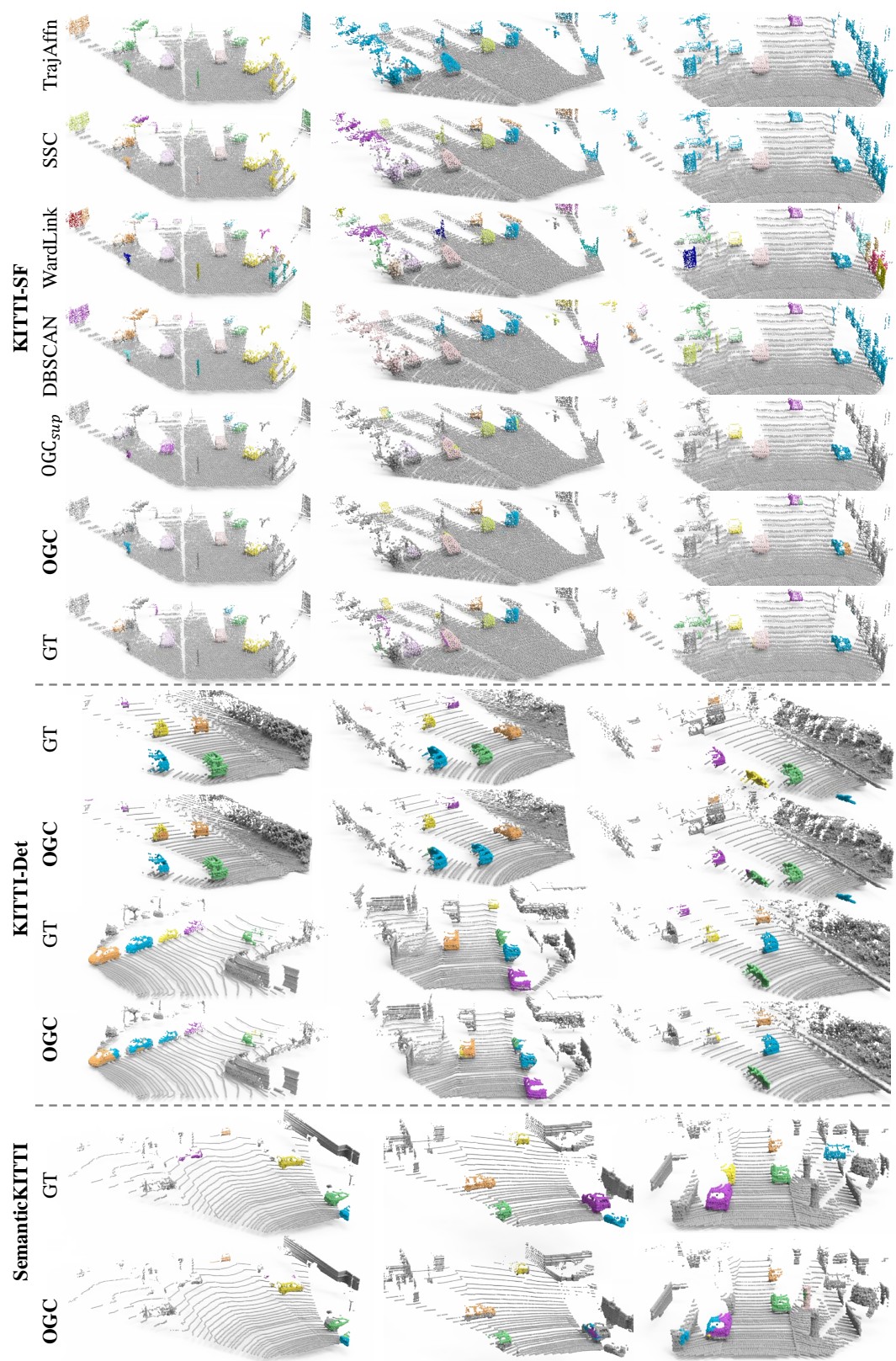

Figure 12: Additional qualitative results on KITTI-SF, KITTI-Det and SemanticKITTI.