# OpenReview forum: "OGC: Unsupervised 3D Object Segmentation from Rigid Dynamics of Point Clouds"
_NeurIPS.cc/2022/Conference — NeurIPS 2022 Accept_

### Official Review · Reviewer_Qxwd · 2022-07-07

**Rating:** 6
**Confidence:** 4
**Soundness:** 2 fair
**Presentation:** 3 good
**Contribution:** 3 good

**Summary:**

This paper introduced a self-supervised approach to do instance segmentation on point clouds.
The key contribution is a set of auxiliary losses to ensure segmentation consistency under per-object rigid transformation and view-variance.
The paper showed large improvement over other unsupervised approaches, which are based on clustering and motion segmentation, and in some cases, the proposed method performs competitively compared to supervised baselines.


---

*I thank the authors for the answer and in particular I appreciate the added experiments. Final recommendation: While I think the proposed method has large potential for improvement, it's relevant to the community as it is the first paper for unsupervised detection. **I maintain my original score.** However the response of the authors provided renewed insights, shedding light to limitations the actual usability due to high training time and the limited generalisability of the learned features. I think it's only fair if the authors mention these in the main paper and link the experiment results to the appendix.*

**Questions:**

- Are $T_k$ and $O_k^t$ jointly optimized?
- What is the training time required to perform the iterative optimization? How does it compare to the supervised version?
- It seems like neither MaskFormer and SceneFlow nets are pretrained. Why? For each dataset, do you need to train these nets from scratch? Do the trained nets to transfer well to another dataset?
- Is the dynamic loss able to recover small objects misclassified as part of another larger object or background? I'm concerned that the estimation $T_k$ will be prone to fit the larger object and ignore the smaller one. This may likely make the segmentation mask fit tighter to the larger object. But I don't see any incentive to segment the smaller object as a new object, say in $o_{n,k'}$

**Limitations:**

Yes

**Strengths And Weaknesses:**

**Strength**
- the first unsupervised method to perform instance segmentation on point clouds (although I'm not a domain expert and couldn't really verify this claim)
- thorough experiments and ablation study on various datasets
- a novel iterative optimization procedure to improve the segmentation mask via improving scene flow

**Weakness**
- The method seems only to be able to detect rigid moving objects, although L155-156 suggests that the geometry invariance loss can also help detect static objects. In general, the object segmentation network doesn't seem to have any fundamental issues in predicting masks for static objects. The dynamic loss also does not penalize static objects (e.g. $M^t = 0, T_k = I$). It would be good if the authors can further clarify the limitation of the proposed method in segmenting static objects.
- I'm not sure the geometry smooth term is well designed. For one, it could lead to blurry prediction boundaries? To me, it makes sense to weigh the smoothness term based on the predicted $T_k$.
- The bold font in a mixed table of supervised and unsupervised comparison is a bit confusing, especially in Table 4, all other methods seem to be better than OGC but OGC is marked bold.
- Also regarding table formatting, it would be good to indicate whether the metric is larger the better or smaller the better, e.g. with $\uparrow$ and $\downarrow$
- One interesting benefit of unsupervised training is that it can be used as a pertaining procedure to boost performance when the data annotation is sparse or for transferring to another domain. The others have shown domain transfer to KITTI-Det, but the baselines are all supervised methods. What's missing is to demonstrate the advantage of the OGC(unsup) as a pertaining step for semi-supervised learning with partial training data, preferably doing so while comparing other unsupervised methods.

---

> ### Author Response · Authors · 2022-08-02
> **Response to Reviewer Qxwd: Questions 1-3**
>
> We appreciate the reviewer's thoughtful comments and address the main concerns below.
>
> **Q1: OGC in segmenting static objects.**
>
> **A1:** During training, our method may not be able to learn from static objects effectively. However, our invariance loss enforces the learned object geometries to be view-invariant. Therefore, if similar objects are moving in other point cloud samples, our method is still likely to learn the geometries of such objects and can successfully segment them during inference.
>
> During inference, our method takes single point cloud frames as input and will not be affected by the motion information. As shown in appendix at Section B.4 (Figure 11), our method OGC can successfully segment static cars parking alongside the road on the KITTI benchmarks.
>
> **Q2: Weight the smoothness term based on the predicted $T_k$.**
>
> **A2:** Weighing by $T_k$ may not be feasible, as the estimated $T_k$ of $K$ objects are the same for all points. We are not sure if the reviewer is suggesting we weight the smoothness term by the fitted rigid motion $\sum_k O^t_k (\boldsymbol{T}_k \circ \boldsymbol{P^t})$. This fitted rigid motion is not reliable, as the inferred object masks $O^t_k$ are meaningless at the initial training stage. Instead, using the given motions $M^t$ to weight the smoothness term might be feasible, as they are fixed during training and assumed to be high-quality.
>
> We evaluate a variant of the smoothness regularization which is weighted by the inter-point motion similarity. Details are in appendix at Section B.3. This variant selectively enforces the object mask smoothness among points with close locations and similar motions. Intuitively, it may avoid blurry predictions on object boundaries, *if the motions are sufficiently accurate*.
>
> The results are shown below, with this motion-similarity-weighted variant denoted by $l_{smooth}'$. We see that $l_{smooth}'$ brings 1.7 higher AP score for our OGC method (Ablation1). However, when the motions are of low quality, $l_{smooth}'$ is likely to introduce additional noise and harm the training. To validate it, we use two groups of distorted scene flows from Section B.1 of appendix to conduct another two groups of ablation studies. Results of Ablations 2/3 show that $l_{smooth}'$ are misled by the distorted scene flows and the network becomes less robust.
>
> Overall, the motion-similarity-weighted smoothness regularization can indeed benefit the segmentation performance on the condition that the estimated motions should be of high quality. To this end, it may be a good strategy to use our vanilla regularizer $l_{smooth}$ in the early round of our iterative optimization, while using the motion-similarity-weighted regularizer $l_{smooth}'$ afterwards.
>
> |       | Flow Source | \#$R$ | Regularizer | AP | PQ | F1 | Pre | Rec | mIoU | RI |
> | - | - | - |  - |  - |  - |  - |  - |  - |  - |  - |
> |Ablation1| FlowStep3D (epoch=120) | 2 | $l_{smooth}$ | 50.1 | 40.5 | 51.1 | 46.5 | 56.8 | 62.0 | 93.1 |
> |               | FlowStep3D (epoch=120) | 2 | $l_{smooth}'$| 51.8 | 40.2 | 49.9 | 44.2 | 57.2 | 62.1 | 93.1 |
> |Ablation2| GT + Gaussian (std=10.0) | 1 | $l_{smooth}$ | 61.4 | 51.3 | 61.7 | 57.2 | 66.8 | 69.4 | 95.0 |
> |               | GT + Gaussian (std=10.0) | 1 | $l_{smooth}'$| 57.5 | 43.4 | 52.2 | 45.0 | 62.1 | 67.7 | 94.4 |
> |Ablation2| GT + Gaussian (std=20.0) | 1 | $l_{smooth}$ | 59.0 | 47.7 | 58.2 | 53.3 | 64.0 | 67.2 | 94.7 |
> |               | GT + Gaussian (std=20.0) | 1 | $l_{smooth}'$| 56.3 | 40.8 | 49.3 | 41.2 | 61.3 | 66.2 | 93.8 |
>
> **Q3: Table formatting issues.**
>
> **A3:** Thanks for pointing out these issues. They have been fixed in the revised paper.

---

> > ### Author Response · Authors · 2022-08-02
> > **Response to Reviewer Qxwd: Questions 4-6**
> >
> > **Q4: OGC as pre-training.**
> >
> > **A4:** Thank you for sharing this insight. We conduct additional experiments to investigate whether our unsupervised method OGC can be used as a pre-training technique before fully-supervised fine-tuning with a small amount of labeled data, like other popular unsupervised representation learning methods. To do this, we firstly keep a 1/10 subset of labelled point clouds from KITTI-Det training set, and then use our OGC model unsupervisedly trained on KITTI-SF to be fine-tuned on the labelled subset. For comparison, we additionally train a new model with full supervision on the same labelled subset from scratch.
> >
> > As shown in the Table below, we can see that: 1) Not surprisingly, using the pre-trained OGC model followed by fine-tuning brings a significant improvement over our unsupervised OGC model, *i.e.*, the 1st row $vs$ the 3rd row. 2) However, the (pre-train + fine-tune) strategy fails to outperform the fully-supervised model from scratch, *i.e.*, the 1st row $vs$ the 2nd row. Fundamentally, this is because our unsupervised OGC is not dedicated to learn general intermediate representations for multiple downstream tasks. Instead, our OGC is task-driven and it aims to directly segment objects from raw point clouds. The learned latent representations in unsupervised training are likely to be different from the latent representations learned in fully-supervised training. In this regard, a na"ive combination of (pre-train + fine-tune) may confuse the network and give inferior results. Nevertheless, how to effectively leverage the unsupervised model along with full supervision is an interesting direction and we leave it for future exploration.
> >
> > | training strategy | AP | PQ | F1 | Pre | Rec | mIoU | RI |
> > | :-: | - | - | - | - | - | - | - |
> > | train OGC$_{sup}$ on 10\% labelled KITTI-Det | 71.5 | 58.3 | 68.4 | 61.8 | 76.7 | 79.1 | 96.0 |
> > |train OGC on unlabelled KITTI-SF + finetune on 10\% labelled KITTI-Det| 66.4 | 53.6 | 63.3 | 56.0 | 72.7 | 76.2 | 95.1 |
> > | train OGC on unlabeled KITTI-SF | 41.0 | 30.9 | 37.7 | 31.4 | 47.0 | 59.9 | 85.0 |
> >
> > **Q5: Are $T_k$ and $O^t_k$ jointly optimized?**
> >
> > **A5:** No, they are optimized in turn, as demonstrated in lines 136-143 of the main paper. First, we fix $O^t_k$ and optimize $T_k$. The Weighted Kabsch algorithm gives a closed-form solution for this optimization step. Second, we fix $T_k$ and optimize $O^t_k$. This optimization step is achieved by training the object segmentation network with gradient descent.
> >
> > **Q6: Time cost of iterative optimization $vs$ supervised training.**
> >
> > **A6:** Each round of the iterative optimization consists of unsupervised object segmentation training and scene flow improvement. The latter is roughly equivalent to a one-go segmentation inference on all samples, and the time consumed is negligible relative to the training. The number of rounds is typically set as 2 in our experiments. Therefore, we compare the total time cost of "2 $\times$ unsupervised segmentation training" to that of the fully-supervised segmentation training. As shown below, although our iterative optimization takes more time, we believe the additional time consumption is much more cost-effective than human annotations.
> >
> > | dataset | iterative optimization (hrs) | fully-supervised training (hrs) |
> > | - |  - | - |
> > | SAPIEN | 16.8 (8.4 $\times$ 2) | 3.8 |
> > | OGC-DR | 29.4 (14.7 $\times$ 2) | 12.4 |
> > | KITTI-SF | 7.8 (3.9 $\times$ 2) | 1.9 |

---

> > > ### Author Response · Authors · 2022-08-02
> > > **Response to Reviewer Qxwd: Questions 7-8**
> > >
> > > **Q7: It seems like neither MaskFormer and SceneFlow nets are pretrained. Why? For each dataset, do you need to train these nets from scratch? Do the trained nets transfer well to another dataset?**
> > >
> > > **A7:** Scene flow estimators can transfer to datasets with similar point densities, scene scales and motion scales. However, the three datasets SAPIEN, OGC-DR and KITTI-SF are significantly different from each other in terms of these properties. Therefore, we train scene flow networks from scratch on SAPIEN/OGC-DR seperately. For KITTI-SF, we use the released FlowStep3D model pre-trained on FlyingThings3D.
> > >
> > > Object segmentation networks can transfer to datasets with similar point densities, scene scales and object categories. Again, SAPIEN, OGC-DR and KITTI-SF are different from each other in terms of these properties. Therefore, we train OGC from scratch on all of them. As we have shown, OGC trained on KITTI-SF can generalize to KITTI-Det and SemanticKITTI. This demonstrates the generalization ability of our OGC, in spite of the different point cloud patterns in KITTI-SF (depth from stereo) and KITTI-Det/SemanticKITTI (Lidar).
> > >
> > > **Q8: Is the dynamic loss able to recover small objects misclassified as part of another larger object or background?**
> > >
> > > **A8:** Yes. For each point $\boldsymbol{p}$, the gradient descent of the dynamic rigid loss can be computed as $\frac{- \partial l_{dynamic}}{\partial o^t_k} = - \frac{2}{K} \|\| \boldsymbol{T}_k \circ \boldsymbol{p}^t - (\boldsymbol{p}^t + \boldsymbol{m}^t) \|\|_2$. This means if a point $p$ is misclassified into a wrong object $k$, the gradient descent on the assignment $o^t_k$ is proportional to the motion fitting error. Therefore, when a small object is misclassied as part of another larger object $k$, all the points on this small object will depress their $o^t_k$ unless this larger object $k$ can well fit the motion of the small object. As the object mask for each point is normalized by softmax, decrease of $o^t_k$ will lead to increase of other $o^t_j (j \neq k)$. As a result, this small object will be gradually pushed into another segment which can better fit its motion. In general, our method do not take an explicit splitting mechanism, but we set the number of object masks $K$ to be large enough for the dataset. In this way, objects will automatically find the proper mask assignment.
> > >
> > > For a practical example, the ratio of point numbers between foreground objects and background in the KITTI-SF dataset is only $\sim$ 0.075:1 on average. However, our method still learns the object segmentation from this dataset successfully.

---

> ### Author Response · Authors · 2022-08-09
> **Response to Reviewer Qxwd's final recommendation**
>
> Thank you for your comments and suggestions.
>
> We have further updated our paper. In lines 358-362 of the main paper (page 9, "Section 5. Conclusion"), we discuss the limitations of our method in terms of time cost and the generalizability of the learned features, with reference to the experiments in appendix at Section B.8 and B.9.

---

### Official Review · Reviewer_8Drm · 2022-07-11

**Rating:** 6
**Confidence:** 4
**Soundness:** 2 fair
**Presentation:** 2 fair
**Contribution:** 2 fair

**Summary:**

The paper proposes a pipeline of 3D object segmentation from point clouds. The key idea is to add several regularization losses to enforce geometry consistency on object motion. The proposed method is evaluated on datasets for object part instance segmentation, general object segmentation, and scene flow estimation, where it achieves competitive performance.

**Questions:**

The reviewer was concerned that the authors made a wild claim “we introduce the first unsupervised multi-object segmentation pipeline on 3D point clouds, without needing any human annotations in training” as this has been previously explored in the scene flow community.

**Limitations:**

The evaluation of the SAPIEN and OGC-DR for this paper seems to be limited as these toy-like datasets have zero background and well-separated objects. The reviewer would expect more exploration of more complicated LIDAR datasets while the paper is currently limited on a small KITTI-SF dataset.

**Strengths And Weaknesses:**

**Originality**: The proposed pipeline is a novel combination of existing techniques. Only limited work has been proposed to address the problem of unsupervised 3D object segmentation. This paper leverages self-supervised scene flow estimation models to obtain initial motion estimation from a pair of point clouds and a neural network to obtain initial object masks, which are then used to supervise the learning of the object segmentation backbone, via introducing regularization techniques to enforce rigid transformation on objects, local smoothness, and invariance on semantic predictions. Each subcomponent by itself is not new, but the reviewer gives the credit to the combination.

**Quality**: The submission is not technically sound. Segmenting 3D objects from the dynamics of point clouds is not a new concept, which has been explored in the scene flow estimation community as the motion segmentation task, e.g., experiments in FlowNet3D [1]. While in this submission, the authors explicitly add an object mask model and combine its prediction with the scene flow prediction to obtain auxiliary supervision signals.  The authors identify the creation of auxiliary supervision signals using motion dynamics as the key challenge and push it as the key contribution. However, the resulting three losses, i.e., the multi-object dynamic rigid consistency loss,  the object shape smoothness prior loss, and the object shape invariance loss are well-known techniques in scene flow estimation and augmentation-based self-supervised learning.
- The multi-object dynamic rigid consistency loss.  The concept has been explored in Rigid3DSceneFlow[2], where they add rigid-motion constraints for both ego-motion and object motion.
- The object shape smoothness prior loss. It shares the same spirit of local smoothness losses that are widely used in the scene flow community, where the key idea is to have a similar motion prediction in nearby points. Here the prior loss is a minor modification by changing the motion prediction to the object mask prediction.
- The object shape invariance loss is highly relevant to the unsupervised semantic segmentation pipeline. The authors resolve the binding problem, e.g., point-to-label, using the Hungarian algorithm between predictions from augmented versions of point clouds and then check the consistency between these predictions. The high-level concept comes from augmentation-based self-supervised learning, still, the proposed loss is somewhat novel.

Overall, the reviewer considers the paper has limited novelty due to the above discussions.


[1] FlowNet3D: Learning Scene Flow in 3D Point Clouds. CVPR 2019

[2] Weakly Supervised Learning of Rigid 3D Scene Flow. CVPR 2021


**Clarity:** The submission has some issues.

It is not clear to the reviewer on the object segmentation network. Is it a supervised model? How do the authors obtain the initial multiple object masks without any human annotations? The authors mentioned the transformer for mask prediction. But it remains unclear to the reviewer how the transformer is trained.

Equation (3) is a wrong notation to ignore the assignment. It didn’t make sense to directly associate two sets of object masks without a `permutation’.

---

> ### Author Response · Authors · 2022-08-02
> **Response to Reviewer 8Drm**
>
> We appreciate the reviewer's thoughtful comments and address the main concerns below.
>
> ## Clarity
> **How do the authors obtain the initial multiple object masks without any human annotations?**
>
> We notice that there may be a misunderstanding, as the reviewer also mentioned "*This paper leverages self-supervised scene flow estimation models to obtain initial motion estimation from a pair of point clouds and a neural network to obtain initial object masks, which are then used to supervise the learning of the object segmentation backbone*" in the **Originality** part.
>
> For clarification,
> * Our OGC method only uses a self-supervised scene flow estimator to produce per-point motion estimates as supervision signals. These signals supervise our object segmentation network from scratch.
> * There is no "neural network to obtain initial object masks". The object masks inferred by our object segmentation network are meaningless at the initial training steps. They gradually become reasonable thanks to our designed object geometry consistency losses.
>
> **Missing "permutation" in Equation (3).**
>
> Thanks for pointing out the flaw. It has been updated in the revised paper.
>
> ## Quality
> **A wild claim of "the first unsupervised multi-object segmentation pipeline on 3D point clouds" as this has been explored in scene flow community as the motion segmentation task.**
>
> For clarification,
> * We are different from motion segmentation methods, like mentioned experiments in FlowNet3D and more recent works[1, 2], which require multiple successive frames as input in both training and testing. Instead, our object segmentation network directly estimates object masks from single frames, and therefore is more flexible and general. Taking a simple example: In appendix at Section B.4 (Figure 11), we show qualitative results about OGC segmenting static objects from single point cloud frames in inference. This is already beyond the scope of motion segmentation.
> * To the best of our knowledge, there's no learning-based work that can segment multiple objects from a single point cloud frame without needing object annotations in training. Classic non-learning based methods such as DBSCAN can work, but significantly worse than our method.
>
> In case we miss concurrent similar works, we are grateful if the reviewer can point out and we will compare and discuss the differences.
>
> **The proposed losses are well-known techniques in prior works.**
>
> For clarification,
> * **Difference of dynamic loss from prior work:** Prior works use the rigid motion constraint to push scene flow to be consistent given object masks. For example, in Rigid3DSceneFlow, the object masks are already estimated by DBSCAN clustering. In contrast, our objective is to learn high-quality masks. The scene flow will be temporally fixed during each round of our object segmentation optimization.
> * **Difference of smoothness loss from prior work:** Unlike the smoothness regularization in scene flow estimation, here we demonstrate its effectiveness for object segmentation. The regularization in scene flow estimation aims to maintain the local spatial structures after warping, while ours aims to overcome the over-segmentation issue from the dynamic loss. The physical meanings are actually different.
> * **Difference of invariance loss from prior work:** The prior unsupervised semantic segmentation works (*e.g.*, PiCIE[3]) uses the invariance to transformations as inductive bias about semantics so that clustering from learned feature representations can be semantically consistent. In contrast, the invariance loss in our method aims to generalize the object segmentation strategy learned from moving objects to similar yet static objects.
>
> Our designed losses are well-motivated and extensively analyzed in ablations. The whole network achieves excellent results on four datasets including indoor and outdoor large-scale ones. These clearly show that our method is technically sound and novel.
>
> ## Limitations
> **Exploration of more complicated LIDAR datasets.**
>
> Thanks for the suggestion. We have conducted additional experiment to generalize our OGC from KITTI-SF to the large-scale Lidar dataset SemanticKITTI. More details are present in appendix at Section B.7. It can be seen that, our method can still achieve excellent segmentation results (**43.3 AP**) thanks to our well-designed object geometry consistency losses.
>
> **References:**
> 1. Stefan Andreas Baur, David Josef Emmerichs, Frank Moosmann, Peter Pinggera, Björn Ommer, and Andreas Geiger. SLIM: Self-Supervised LiDAR Scene Flow and Motion Segmentation. ICCV, 2021.
> 2. Jiadai Sun, Yuchao Dai, Xianjing Zhang, Jintao Xu, Rui Ai, Weihao Gu, and Xieyuanli Chen. Efficient Spatial-Temporal Information Fusion for LiDAR-Based 3D Moving Object Segmentation. IROS, 2022.
> 3. Jang Hyun Cho, Utkarsh Mall, Kavita Bala, and Bharath Hariharan. PiCIE: Unsupervised Semantic Segmentation using Invariance and Equivariance in Clustering. CVPR, 2021.

---

> ### Author Response · Authors · 2022-08-09
> **Waiting for Discussion**
>
> Dear reviwer 8Drm,
>
> Since the Author- Reviewer Discussion is closing very soon, we are still waiting for your thoughts on our rebuttal materals (detailed explanation + revisions of the main paper and appendix).
>
> Regarding all your concerns including the technique novelty, writing clarity, and more experiments, we believe they are all clearly addressed and also highlighted by yellow color in the main paper.
>
> We are grateful if you could share your further feedback. Thank you for your time.
>
> Regards,
> Authors

---

> > ### Comment · Reviewer_8Drm · 2022-08-10
> > **Happy to raise the score**
> >
> > Dear authors,
> >
> > The reviewer is happy that the authors have addressed most of the initial concerns and improved the writing clarity. I do not have further questions at the current stage. I have increased the score accordingly.
> >
> > Best,
> > Reviewer 8Drm

---

> > > ### Author Response · Authors · 2022-08-10
> > > **Thanks**
> > >
> > > Dear reviwer 8Drm,
> > >
> > > We really appreciate your very encouraging feedback.
> > >
> > > Regards,
> > > Authors

---

### Official Review · Reviewer_eGGN · 2022-07-11

**Rating:** 6
**Confidence:** 5
**Soundness:** 3 good
**Presentation:** 3 good
**Contribution:** 3 good

**Summary:**

This paper studies the problem of unsupervised 3D object segmentation from a point cloud. In particular, the proposed approach learns the scene flow in a self-supervised manner which is used as supervision signal and enforces the object geometric consistency for segmenting the static objects. The proposed unsupervised approach is evaluated on various synthetic and real dataset to demonstrate the performance.

**Questions:**

1)	How is the MaskFormer [8] trained? Is a pre-trained model applied to obtain the coarse object mask?
2)	Please analyse Influence of the smoothness parameter H for over segmentation.
3)	Could the method handle incomplete point cloud?
4)	It would be great to show the proposed method can handle static objects with qualitative results.

5)	Based on the reviewer’s understanding, the proposed approach mainly handle foreground objects without road. A large number of points belongs to floor or road which can be easily removed by plane fitting. The baseline clustering results should become much better.


**Ethics Review Area:**

["I don’t know"]

**Limitations:**

The paper has discussed limitations of the proposed method.

**Strengths And Weaknesses:**

# Strengths
1)	Unsupervised 3D object segmentation from point cloud is overlooked in the computer vision field. The paper proposed a solution to solve the problem.
2)	Point cloud motion is applied as supervision signal for 3D object point segmentation. The motion consistency indeed provides strong information for 3D object point segmentation.
3)	The geometry invariance loss is applied to encourage the segmentation of static objects in the scene.
4)  The method is evaluated on both synthetic and real datasets.

# Weaknesses
1)	The training loss quality will be influenced by the self-supervised scene flow estimation approach.
2)	The proposed method may fail if two objects are close with similar motion.
3)	There are still over-segmentation issues in the results.
4)   The proposed method can only handle objects from a finite dataset which hinders its generalisation ability.

---

> ### Author Response · Authors · 2022-08-02
> **Response to Reviewer eGGN: Questions 1-4**
>
> We appreciate the reviewer's thoughtful comments and address the main concerns below.
>
> **Q1: Is MaskFormer [8] a pre-trained model offering coarse object mask?**
>
> **A1:** No. The whole object segmentation network is end-to-end trained by our object consistency losses from scratch, without any pre-training. There is no coarse object mask at all, because we do not have any annotations. The inferred object masks are meaningless at the very beginning at training stage.
>
> We have clarified all these points in the revised paper.
>
> **Q2: Over-segmentation issues & Influence of the smoothness regularization hyperparameters**
>
> **A2:** Over- and/or under-segmentation issues are the core problems for general object segmentation methods, including fully-supervised, weakly-supervised, and unsupervised schemes.
>
> In our OGC method, the trade-off between over- and under-segmentation is largely balanced by the smoothness regularization loss. To extensively study this issue, we conduct additional ablation studies with regard to different hyperparameters of smoothness loss in appendix at Section B.2 "Choice of Hyperparameters for Smoothness Regularization".
>
> As shown in the Table below, when we strengthen the regularization by enforcing smoothness in a larger local neighborhood (i.e., ablation $H_3$), the Precision score improves with less over-segmentation, while the Recall score is sacrificed. Figure 10 in appendix shows qualitative results. As expected, such hyperparameters control the trade-off between over- and under-segmentation issues.
>
> In general, our method can obtain satisfactory AP scores given a wide range of choices for neighbouring points. Nevertheless, we believe that more advanced designs, such as leveraging spatial connectivity or color cues (if available), may further alleviate this issue.
>
> |          | $(k_1, r_1)$,  $(k_2, r_2)$ |  AP  |  PQ  |  F1  | *Pre* | *Rec* | mIoU | RI |
> |      :-:   |                     -                     |    -    |    -    |    -   |   -   |    -    |    -    |   -   |
> |$H_1$| (4, 0.01), (8, 0.02)             | 90.9 | 82.5 | 87.3 | *82.3* | *92.9* | 89.8 | 97.3 |
> |$H_2$ **(Full OGC)**| (8, 0.02), (16, 0.04)            | 91.3 | 83.5 | 88.5| *84.9* | *92.5* |  89.2  |97.2|
> |$H_3$| (32, 0.08), (64, 0.16)          | 81.7 | 79.6 | 86.4| *91.0* | *82.3* |  79.7  |96.2|
>
> **Q3: Can OGC handle incomplete point clouds?**
>
> **A3:** The overall answer is Yes. The point clouds in the outdoor datasets we used (KITTI-SF, KITTI-Det, SemanticKITTI) all come from single-view scans, thus being incomplete due to self- or cross-occlusions. As shown in Table 3/4/5 in the main paper, our method can successfully handle these datasets.
>
> Moreover, we simulate single-view scans on the mesh models of our OGC-DR indoor dataset and build another Single-View OGC-DR (OGC-DRSV) dataset with incomplete point clouds. As shown below, the performance of our method on OGC-DRSV is still satisfactory, being very close to the supervised baseline. More details are provided in appendix at Section B.6.
>
> |  |  | AP | PQ | F1 | Pre | Rec | mIoU | RI |
> |  - | :-: |   -    |    -    |    -   |   -   |    -    |    -    |   -   |
> | Supervised | OGC$_{sup}$ | 86.1 | 77.3 | 83.9 | 80.9 | 87.1 | 82.9 | 96.5 |
> | Unsupervised | **OGC** | 86.3 | 75.4 | 82.6 | 76.3 | 90.0 | 83.7 | 94.6 |
>
> **Q4: Qualitative results of OGC segmenting static objects.**
>
> **A4:** On KITTI-SF and KITTI-Det datasets, We project the segmented point clouds onto corresponding RGB images for a better visualization, as shown in appendix at Section B.4, Figure 11.

---

> > ### Author Response · Authors · 2022-08-02
> > **Response to Reviewer eGGN: Questions 5-6**
> >
> > **Q5: Improve the baselines on KITTI-SF.**
> >
> > **A5:** Thank you for the great suggestion. We leverage the prior about ground planes in the KITTI-SF dataset to improve baseline methods. First, we detect and temporarily remove the ground plane, letting the baseline algorithms to segment above-ground points only. For TrajAffn and SSC, we can use motion information to merge the ground points with above-ground segments that are likely to be part of the static background. To do this, we employ the Kabsch algorithm to estimate the rigid transformation of the ground. Then the above-ground segments whose motions are well fitted by the ground's transformation will be incorporated. For WardLinkage and DBSCAN, the ground is treated as a separate segment.
> >
> > As shown in the results below, SSC and WardLinkage got remarkable improvements. For TrajAffn and DBSCAN, although the quantitative performance gain is not significant, we find their qualitative results become more meaningful as shown in Figure 12 in appendix. Therefore, we take the improved versions of baselines for comparison to OGC. Our method still holds significant advantages over the baselines. The quantitative and qualitative comparison in Table 3 and Figure 3 of the main paper have been updated.
> >
> > ($^{\dagger}$ denotes the improved versions with the ground points specially handled)
> > |  |  | AP | PQ | F1 | Pre | Rec | mIoU | RI |
> > | - | - | - | - | - | - | - | - | - |
> > |Unsupervised Motion Segmentation| TrajAffn | 22.4 | 34.7 | 42.8 | 40.7 | 45.2 | 48.8 | 83.5 |
> > | | TrajAffn$^{\dagger}$ | 23.2 | 29.3 | 41.7 | 35.8 | 50.0 | 47.9 | 58.3 |
> > | | SSC | 0.6 | 5.1 | 7.5 | 6.2 | 9.4 | 19.3 | 33.2 |
> > | | SSC$^{\dagger}$ | 12.2 | 19.9 | 27.7 | 21.9 | 37.4 | 41.3 | 48.5 |
> > |Unsupervised Methods | WardLinkage | 0.4 | 2.4 | 3.8 | 2.2 | 14.4 | 26.9 | 15.8 |
> > | | WardLinkage$^\dagger$ | 13.1 | 16.3 | 22.9 | 13.7 | **69.8** | 60.5 | 44.9 |
> > | | DBSCAN | 12.6 | 29.8 | 32.8 | 46.5 | 25.3 | 31.1 | 84.7 |
> > | | DBSCAN$^\dagger$ | 13.1 | 22.8 | 32.6 | 26.7 | 42.0 | 42.6 | 55.3 |
> > | | **OGC(Ours)** | **50.1** | **40.5** | **51.1** | **46.5** | 56.8 | **62.0** | **93.1**
> >
> > **Q6: Training loss quality is influenced by scene flow estimator.**
> >
> > **A6:** Thank you for sharing this insight. We systematically analyze the robustness of our OGC by simulating different types of scene flow distortions on the OGC-DR and KITTI-SF datasets. To do this, we conduct experiments on three types of distorted scene flows:
> > * (i) We add different degrees of zero-mean Gaussian noise into the ground truth scene flows, and these noisy scene flows are used to supervised our object segmentation network.
> > * (ii) We use insufficiently trained scene flow estimators to produce low-quality scene flow estimations for supervision.
> >  * (iii) We use the initial scene flow estimations which has not been refined by our iterative optimization.
> >
> > On our OGC-DR dataset, we evaluate all these ablations. On the KITTI-SF dataset, we only evaluate (i) and (iii), as we use a FlowStep3D model publicly released by the authors to estimate scene flows for KITTI-SF. The intermediate training models are not available to evaluate (ii).
> >
> > All results are available in appendix at Section B.1 "Robustness to Scene Flow Distortions". From the results, we can see that:
> > * Our OGC method has strong robustness to Gaussian noise on both OGC-DR and KITTI-SF datasets. This is because the Weighted Kabsch algorithm inside our dynamic rigid loss inherently smooths the Gaussian-like noise in scene flows.
> > * In contrast, the inaccurate scene flows estimated by the network indeed incur a drop in the object segmentation performance. However, given more rounds in our iterative optimization, *e.g.*, \#R$\geq$ 2, the overall performance is still satisfactory. Basically, this is because our object-aware ICP is designed to correct such inconsistency (biases) in the flows.

---

> > > ### Author Response · Authors · 2022-08-02
> > > **Response to Reviewer eGGN: Questions 7-8**
> > >
> > > **Q7: The proposed method may fail if two objects are close with similar motion.**
> > >
> > > **A7:** We need to discuss this issue in the training and inference phases respectively.
> > > * During training, if there are two close objects, A and B, with similar motions in the same point cloud, our method may not be able to learn from this sample effectively. Nevertheless, our invariance loss enforces the learned object geometries to be view-invariant. Therefore, if there are candidate objects similar to A (or B) independently moving in other point clouds, our method still learns the geometries of such objects and is likely  successful to segment A (or B) during inference.
> > > * During inference, our method takes a single-frame point cloud as input and will not be affected by motion information. Taking a simple example: multiple static cars and background are spatially adjacent and have similar motions (static), and our method can still separate them. Qualitative examples are provided in appendix at Section B.4, Figure 11.
> > >
> > > **Q8: The proposed method can only handle objects from a finite dataset which hinders its generalisation ability.**
> > >
> > > **A8:** In fact, our method has demonstrated similar generalization abilities to fully-supervised methods. For example,
> > > * **Generalization to novel object instance**: We generalize to the OGC-DR testing set containing different instances from the training set, though the object categories are the same.
> > > * **Generalization from KITTI-SF to KITTI-Det**: As shown in Table 4 of Section 4.3, our method obtains excellent results on KITTI-Det.
> > > * **Generalization from KITTI-SF to SemanticKITTI**: As shown in Table 5 of Section 4.4, our method can still successfully segment cars and trucks on the large-scale Lidar point clouds in SemanticKITTI.
> > >
> > > Nevertheless, segmenting objects of never-seen categories in inference is unlikely possible for our method. In fact, this is also not feasible for fully-supervised methods, fundamentally because learning novel object category geometry involves more advanced zero-shot learning techniques, which is beyond this paper.

---

### Official Review · Reviewer_dUuM · 2022-07-13

**Rating:** 8
**Confidence:** 4
**Soundness:** 3 good
**Presentation:** 3 good
**Contribution:** 3 good

**Summary:**

This paper proposed a 3D object segmentation method that can be trained in an unsupervised fashion. The core underlying assumption is the rigid motion dynamics and shape invariance between point cloud sequences. By carefully designing geometry consistency losses, supervision signals can be effectively extracted from a pair of point clouds. The paper has demonstrated promising results on multiple different datasets and scenarios.

**Questions:**

1) Motion rigidity is an excellent clue for learning object segmentation. However, in certain datasets, or for certain sensor modality, the actual shape invariance and rigid dynamic might not hold in the sensory data. For example, rolling shutter effect is a very well known effect that distorts high speed moving objects captured by a rolling shutter camera. For point clouds captured by mechanical lidar, motion distortion caused by scene flow, especially high speed moving objects, such as vehicles coming from the opposite direction, is usually very obvious. This paper has made great progress in leveraging object rigidity, but it would be interesting to see, how different degrees of motions can affect the proposed method. Especially, for cases like mechanical lidar point cloud, if motion distortion can affect the proposed method.

**Limitations:**

1) The proposed method relies on the object rigidity for object segmentation. But there are objects, such as articulated buses, semi-truck with trailers, that changes shapes during their movement. The proposed method might not be able to handle such cases well.

**Strengths And Weaknesses:**

1) This paper carefully designed multiple loss functions, which powers the unsupervised learning of object segmentation. Removing the need for large scale human annotation is a critical step towards making such method applicable in different new fields and settings.
2) This paper demonstrated great results on multiple datasets.

---

> ### Author Response · Authors · 2022-08-02
> **Response to Reviewer dUuM**
>
> We appreciate the reviewer's thoughtful comments and address the main concerns below.
>
> **Q1: How can motion (scene flow) distortions affect OGC?**
>
> **A1:** Thank you for sharing this insight. We agree that the motion distortion is an important issue in real-world applications. Since we do not have devices to collect mechanical Lidar point clouds, we turn to systematically analyze the robustness of our OGC by simulating different types of scene flow distortions on the OGC-DR and KITTI-SF datasets. To do this, we conduct experiments on three types of distorted scene flows:
> * (i) We add different degrees of zero-mean Gaussian noise into the ground truth scene flows, and these noisy scene flows are used to supervised our object segmentation network.
> * (ii) We use insufficiently trained scene flow estimators to produce low-quality scene flow estimations for supervision.
> * (iii) We use the initial scene flow estimations which has not been refined by our iterative optimization.
>
> On our OGC-DR dataset, we evaluate all these ablations. On the KITTI-SF dataset, we only evaluate (i) and (iii), as we use a FlowStep3D model publicly released by the authors to estimate scene flows for KITTI-SF. The intermediate training models are not available to evaluate (ii).
>
> All results are available in appendix at Section B.1 "Robustness to Scene Flow Distortions". From the results, we can see that:
> * Our OGC method has strong robustness to Gaussian noise on both OGC-DR and KITTI-SF datasets. This is because the Weighted Kabsch algorithm inside our dynamic rigid loss inherently smooths the Gaussian-like noise in scene flows.
> * In contrast, the inaccurate scene flows estimated by the network indeed incur a drop in the object segmentation performance. However, given more rounds in our iterative optimization, *e.g.*, \#R$\geq$ 2, the overall performance is still satisfactory. Basically, this is because our object-aware ICP is designed to correct such inconsistency (biases) in the flows.
>
> **Q2: There are objects, such as articulated buses, semi-truck with trailers, that changes shapes during their movement. The proposed method might not be able to handle such cases well.**
>
> **A2:** We agree that such non-rigid cases indeed cannot be handled by our OGC appropriately. For clarity, we explicitly highlight that our paper focuses on the rigid object segmentation instead of non-rigid or deformable cases. (See lines 78-81 in page 2 of Introduction, main paper).

---

### Author Response · Authors · 2022-08-02
**Overall Response**

We appreciate all valuable comments. After carefully improving the quality of our submission, we present here a revised paper together with the supplementary material. Changes in the main paper have been highlighted and include:
* Clarification of the difference from prior works and the scope
* Clarification of our object segmentation network
* Improved baselines on KITTI-SF
* Additional evaluation on the SemanticKITTI dataset

Specifically, additional experiments and analysis to address the reviewers' concerns are collected into the supplementary material at Section B "Appendix for Rebuttal" (Page 24-29) and include:
* **B.1** Robustness to Scene Flow Distortions
* **B.2** Choice of Hyperparameters for Smoothness Regularization
* **B.3** Weighted Smoothness Regularization via Motion Similarity
* **B.4** Qualitative Results for Static Object Segmentation
* **B.5** Improved Baselines on KITTI-SF (**added into main paper**)
* **B.6** Evaluation on Incomplete Point Cloud Dataset OGC-DRSV
* **B.7** Evaluation on SemanticKITTI (**added into main paper**)
* **B.8** OGC as Pre-training
* **B.9** Time Cost of OGC vs. Supervised Training

---

### Meta-Review · Area_Chair_zTQQ · 2022-08-25

**Recommendation:** Accept
**Confidence:** Certain

**Metareview:**

This is an interesting paper that proposes a novel unsupervised approach for object segmentation from point clouds, for rigid objects. The strong results demonstrated can be impactful both for 3d as well as potentially 2d vision. After rebuttal, all 4 expert reviewers are convinced that the paper should be accepted, so the decision to accept the paper was easy.

**Award:**

No

---

### Decision · Program_Chairs · 2022-09-14

Accept